# The VIP-VPAC2 neuropeptidergic axis is a cellular pacemaking hub of the suprachiasmatic nucleus circadian circuit

Andrew P. Patton [1,5], Mathew D. Edwards[1,2,5], Nicola J. Smyllie [1,5], Ryan Hamnett [1,3,5], Johanna E. Chesham [1], Marco Brancaccio [1,4], Elizabeth S. Maywood[1] & Michael H. Hastings [1✉]

The hypothalamic suprachiasmatic nuclei (SCN) are the principal mammalian circadian timekeeper, co-ordinating organism-wide daily and seasonal rhythms. To achieve this, cell-autonomous circadian timing by the ~20,000 SCN cells is welded into a tight circuit-wide ensemble oscillation. This creates essential, network-level emergent properties of precise, high-amplitude oscillation with tightly defined ensemble period and phase. Although synchronised, regional cell groups exhibit differentially phased activity, creating stereotypical spatiotemporal circadian waves of cellular activation across the circuit. The cellular circuit pacemaking components that generate these critical emergent properties are unknown. Using intersectional genetics and real-time imaging, we show that SCN cells expressing vasoactive intestinal polypeptide (VIP) or its cognate receptor, VPAC2, are neurochemically and electrophysiologically distinct, but together they control de novo rhythmicity, setting ensemble period and phase with circuit-level spatiotemporal complexity. The VIP/VPAC2 cellular axis is therefore a neurochemically and topologically specific pacemaker hub that determines the emergent properties of the SCN timekeeper.

[1] MRC Laboratory of Molecular Biology, Francis Crick Ave., Cambridge Biomedical Campus, Cambridge CB2 0QH, UK. [2] Present address: UCL Sainsbury Wellcome Centre for Neural Circuits and Behaviour, London, UK. [3] Present address: Department of Neurosurgery, Stanford University, Stanford, USA. [4] Present address: Department of Brain Sciences, UK Dementia Research Institute, Imperial College London, London, UK. [5] These authors contributed equally: Andrew P. Patton, Mathew D. Edwards, Nicola J. Smyllie, Ryan Hamnett. ✉email: mha@mrc-lmb.cam.ac.uk

Circadian clocks enable organisms to anticipate and adapt to daily and seasonal environments[1]. The principal circadian clock in mammals is the retinorecipient, hypothalamic suprachiasmatic nucleus (SCN)[2]. It generates a molecular representation of solar time via a cell-autonomous transcriptional/translational negative feedback loop (TTFL), whereby self-sustained circadian oscillations in expression of the negative transcriptional regulators Period (Per) and Cryptochrome (Cry) are driven by heterodimers of the positive regulators Clock and Bmal1[3]. Almost all cells of the body contain similar TTFLs, but rely on daily systemic cues controlled by the SCN to maintain their oscillation[4] and to synchronise them with solar time[5]. As the central circadian co-ordinator, the SCN is unique in its intrinsic ability to generate robust and precise circadian oscillations of TTFL-dependent gene expression and electrophysiological activity, even in the absence of environmental cues. These autonomous oscillations are so powerful that they can persist indefinitely in vivo and ex vivo in organotypic slice culture[6,7].

This robustness and precision of the SCN are emergent (i.e. circuit-level) properties arising from cellular interactions. When the network is compromised pharmacologically[8] or by dispersal in culture[9], the individual cells lose circadian coherence and are more sensitive to genetic perturbation. In the intact circuit, however, robustness is accompanied by cellular synchrony across the SCN, with tightly defined ensemble period and ensemble phase. Importantly, the phases of cellular TTFLs across the SCN are staggered as localised cell groups exhibit peak circadian activities in a distinct sequence. This generates stereotypical spatio-temporal waves of gene expression and intracellular calcium levels ($[Ca^{2+}]_i$) that progress from the dorsomedial to ventral SCN[10]. These waves are thought to be important in allowing the SCN to broadcast distinct and appropriate circadian signals to various targets in the brain[2] and also for circuit-level encoding of daylength, whereby the SCN cues seasonal rhythms[11,12].

Its emergent properties are critical to the paramount circadian role of the SCN, but the cellular circuit elements and functions that create them are not known[2]. The SCN consists of two subdivisions. The core contains cells that express vasoactive intestinal polypeptide (VIP) or gastrin releasing peptide (GRP), and is the entry point for retinal entrainment[13], while the shell contains cells expressing arginine-vasopressin (AVP) and the VIP receptor VPAC2. Loss of genes encoding VIP or VPAC2 compromises cellular synchrony and circadian amplitude[14], while direct optogenetic activation of VIP cells is sufficient to reset the ensemble phase of the SCN[13], likely via VIP-dependent changes in electrical activity and/or gene expression in VPAC2-expressing cells[15]. In addition, cells expressing neuromedin S (NMS) and D1a dopamine receptors (DRD1A) extend across core and shell[16,17], while single-cell transcriptomics has revealed further cellular heterogeneity, highlighting potential topological features of the SCN circuit that include several signalling axes, each consisting of neuropeptide- and receptor-expressing cellular populations[18]. Intersectional genetics has recently been used in an attempt to determine the particular contributions of these defined cellular elements to SCN network properties. From this, it is clear that the cell-autonomous circadian properties of NMS- and DRD1A-expressing neurons[16,17] are important determinants of SCN circuit functions but, paradoxically, signalling by NMS or dopamine is not necessary for robust time-keeping. The contributions of these cells (or sub-groups of them) to circuit functions, therefore, is likely mediated by other signalling factors, which may also include astrocyte-derived signals[19]. Thus, a consensus view on the cellular, neurochemical and topological origin of network properties of the SCN remains elusive. To put it simply, which cell population(s) constituting these signalling axes are the SCN pacemakers? To address this, we focussed on the cells of the neurochemically explicit VIP/VPAC2 axis for reasons outlined above[2,13], and the observation that paracrine VIP-ergic signalling can re-programme circuit function[14,15,20]. To what extent do the cell-autonomous circadian properties of VIP and VPAC2 cells, alone or in combination, determine the circuit-level ensemble properties of the SCN? Our results show that the VIP/VPAC2 cellular axis is a neurochemically and topologically specific pacemaker hub that determines the emergent properties of the SCN timekeeper.

## Results

**VIP/VPAC2 cells contribute to SCN spatiotemporal complexity**. We characterised mouse lines expressing Cre-recombinase in either VIP or VPAC2 cells, mapping cellular distributions by inter-crossing with R26R-EYFP genomic reporter mice. EYFP-labelled cells exhibited contrasting SCN distributions: VIP^EYFP cells clustered ventrally within the core whereas VPAC2^EYFP cells localised around the dorsal, lateral and medial shell (Supplementary Fig. 1a, b). Immunostaining and in situ hybridisation of SCN sections showed clear segregation of VIP and VPAC2 (or AVP) cells (Supplementary Fig. 1c–f), with minimal evidence for mutual VIP–VIP cell signalling via VPAC2. These lines therefore provided selective genetic access to distinct VIP or VPAC2 cellular compartments.

The emergent network properties of the SCN are evident in CCD recordings of Per2::Luciferase bioluminescence in organotypic slices. These reveal striking spatio-temporal waves whereby the SCN can be mapped into ~6 temporally defined, regionally specific phase-clusters (Fig. 1a, Supplementary Movie 1, Supplementary Fig. 2). This phase-map shows earlier activity in the extreme dorsomedial lip (region 1, cells defined, at least in part, by RGS16 expression[21]) progressing into the ventral core before spreading into the dorsal, lateral and medial SCN shell. Regional distribution of VIP^Cre and VPAC2^Cre cells, as revealed by Cre-dependent EYFP fluorescence (delivered by adeno-associated virus (AAV)), mapped to different phase clusters: VIP^EYFP neurons were within the ventral phase-advanced core, straddling regions 2 to 4, and with a ventral-to-dorsal wave progression (Fig. 1a, Supplementary Fig. 3a). Conversely, the VPAC2^EYFP neurons mapped dorsally across the shell, straddling delayed phase regions 4–6, with a predominantly lateral wave progression (Fig. 1a, Supplementary Fig. 3b). This coarse mapping suggested that the two cell types are differentially phased with contrasting spatiotemporal dynamics of circadian activation. To examine this directly, we monitored the circadian cytosolic $[Ca^{2+}]_i$ rhythms of VIP and VPAC2 cells as reported by GCaMP6f[6,22,23]. When expressed pan-neuronally (via AAV under the synapsin promoter), GCaMP6f reported strong circadian oscillations across the SCN peaking at ca. CT7 (CT12 is the peak of the accompanying Per2::Luciferase signal), and with a spatio-temporal wave comparable to Per2::Luciferase: emerging in the dorsal SCN, entering the ventral core and with a final dorsal, lateral and medial spread into the SCN shell (Supplementary Fig. 4a–e). Cre-conditionally expressed GCaMP6f reported strong $[Ca^{2+}]_i$ rhythms in both VIP^GCaMP6f and VPAC2^GCaMP6f cells (Fig. 1b–d), but with different waveforms: VIP^GCaMP6f oscillations showed a sharper rise and broader peak than VPAC2^GCaMP6f cells (Fig. 1e), indicative of population-specific cellular properties. Importantly, when registered against the Per2::Luciferase rhythm, the cycle of VIP^GCaMP6f cells was significantly phase-advanced relative to pan-neuronal and VPAC2^GCaMP6f oscillations (Fig. 1e, f) (Neuronal: CT 7.6 ± 0.2 h; VIP: CT6.1 ± 0.4 h; VPAC2: CT7.8 ± 0.2 h). Consistent with this, when mapped in SCN space, the

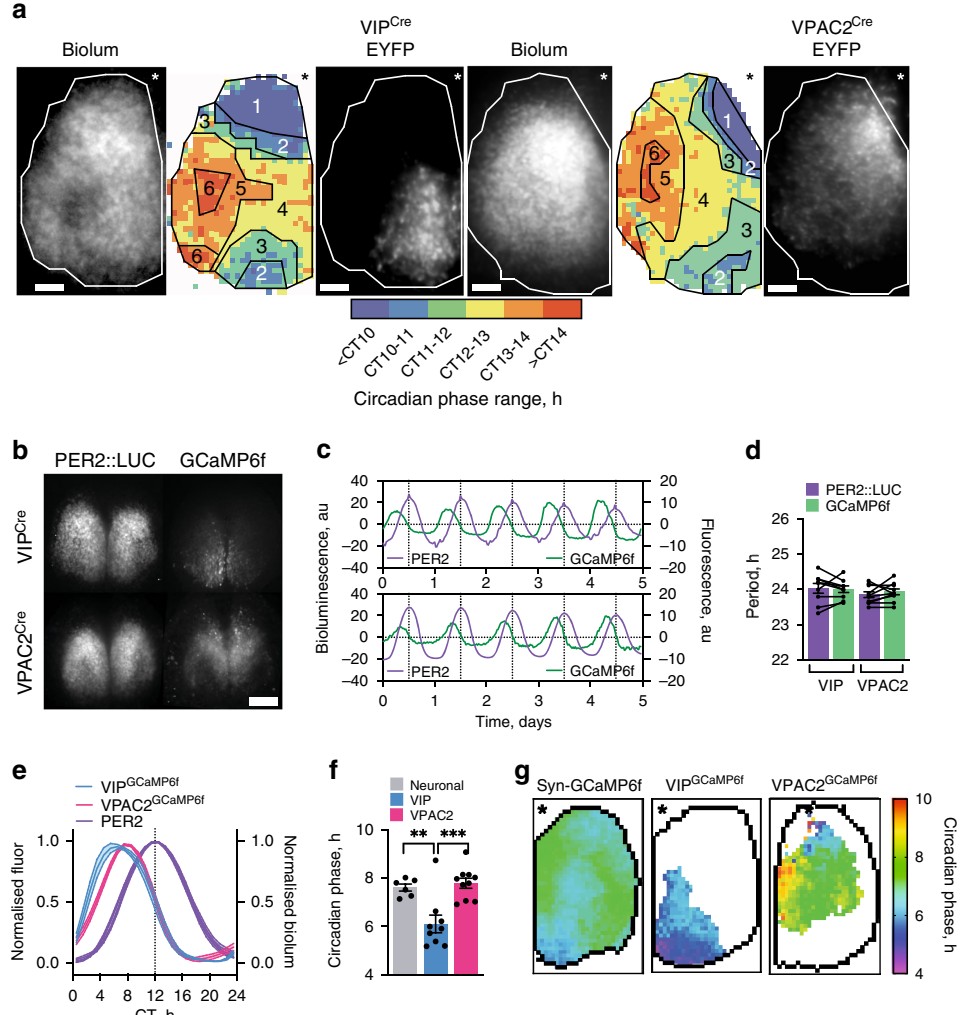

**Fig. 1 Regional distribution and circadian phase of VIP$^{Cre}$ and VPAC2$^{Cre}$ cells in SCN. a** Registration of VIP$^{Cre}$ and VPAC2$^{Cre}$ cells to corresponding phase maps of SCN spatiotemporal bioluminescent wave. For both cell types, left panel shows projections of average bioluminescence emissions recorded by CCD; middle panel shows the corresponding phase map categorically coloured as 1-h phase bins overlaid with manually annotated sequential phase clusters; and right panel shows distribution of Cre-expressing cells revealed by Cre-conditional EYFP intensity. The perimeter of the SCN is delineated by the white outline, and the location of the third ventricle is denoted by asterisks. Note registration of VIP$^{Cre}$ and VPAC2$^{Cre}$ cells with earlier and later phase clusters, respectively. These images are representative of a set of three experiments for VIP$^{EYFP}$ and VPAC2$^{EYFP}$ each, which are presented in Supplementary Fig. 3. Scale bars = 50 μm. **b** Representative micrographs of Per2::Luciferase in register with GCaMP6f conditionally targetted to VIP (upper) or VPAC2 (lower) cells. Scale bars = 250 μm. **c** Representative detrended Per2::Luciferase (purple) and conditionally targetted GCaMP6f traces (green) from recordings from VIP$^{Cre}$ (upper) and VPAC2$^{Cre}$ (lower) slices. **d** Paired period measures of Per2::Luciferase (purple) and conditional GCaMP6f (green) rhythms. **e** Phase-aligned normalised single cycles of Per2::Luciferase (two overlaid purple) versus corresponding VIP$^{GCaMP6f}$ (blue) and VPAC2$^{GCaMP6f}$ (pink) cycles. Lines and shading are mean ± SEM. **f** Circadian-normalised phase measures for pan-neuronal GCaMP6f (grey), VIP$^{GCaMP6f}$ (blue) and VPAC2$^{GCaMP6f}$ (pink), (**$p = 0.004$, ***$p = 0.0005$). **g** Circadian-normalised phase-maps of neuronal GCaMP6f (left), VIP$^{GCaMP6f}$ (middle) and VPAC2$^{GCaMP6f}$ (right). SCN outlines from corresponding Per2::Luciferase phase-maps, asterisks mark third ventricle. In all histograms, individual points represent individual SCN slices and bars are mean ± SEM. Statistics: **d** paired two-tailed $t$-tests; **f** ordinary one-way ANOVA with Holm-Sidak's correction for multiple comparisons. For GCaMP6f imaging (**d-f**): $n = 6$, 9 and 10 for neuronal, VIP$^{GCaMP6f}$ and VPAC2$^{GCaMP6f}$. Only significant comparisons ($p < 0.05$) are shown. Source data are provided as a source data file.

VIP$^{GCaMP6f}$ and VPAC2$^{GCaMP6f}$ cells occupied distinct cell-type-specific phase-territories relative to whole-field $[Ca^{2+}]_i$ and Per2::Luciferase bioluminescence, with VIP early and VPAC2 late (Fig. 1g, Supplementary Fig. 4f, g). Furthermore, within these domains they exhibited contrasting local spatio-temporal dynamics: VIP$^{GCaMP6f}$ neurons displayed a strong ventral-to-dorsal wave, whereas VPAC2$^{GCaMP6f}$ neurons displayed two features: an early rise in a small number of dorsomedial lip neurons, followed by the bulk of the wave emanating from the centre of the slice medially and laterally (Fig. 1g, Supplementary Fig. 4e–g). The SCN-wide emergent wave of $[Ca^{2+}]_i$ therefore contains locally specific sub-waves attributable to VIP and VPAC2 cells.

**VIP/VPAC2 cells are electrophysiologically distinct**. Circadian cytosolic $[Ca^{2+}]_i$ rhythms reflect, in part, changes in neuronal firing rate. We therefore tested directly whether VIP and VPAC2 cells have differentially phased electrical rhythms and thus the potential to encode separate temporal elements of the SCN cycle. Untargetted electrophysiological recordings from SCN slices phase mapped to ensemble bioluminescence rhythms revealed circadian changes in spontaneous firing rate (SFR), varying

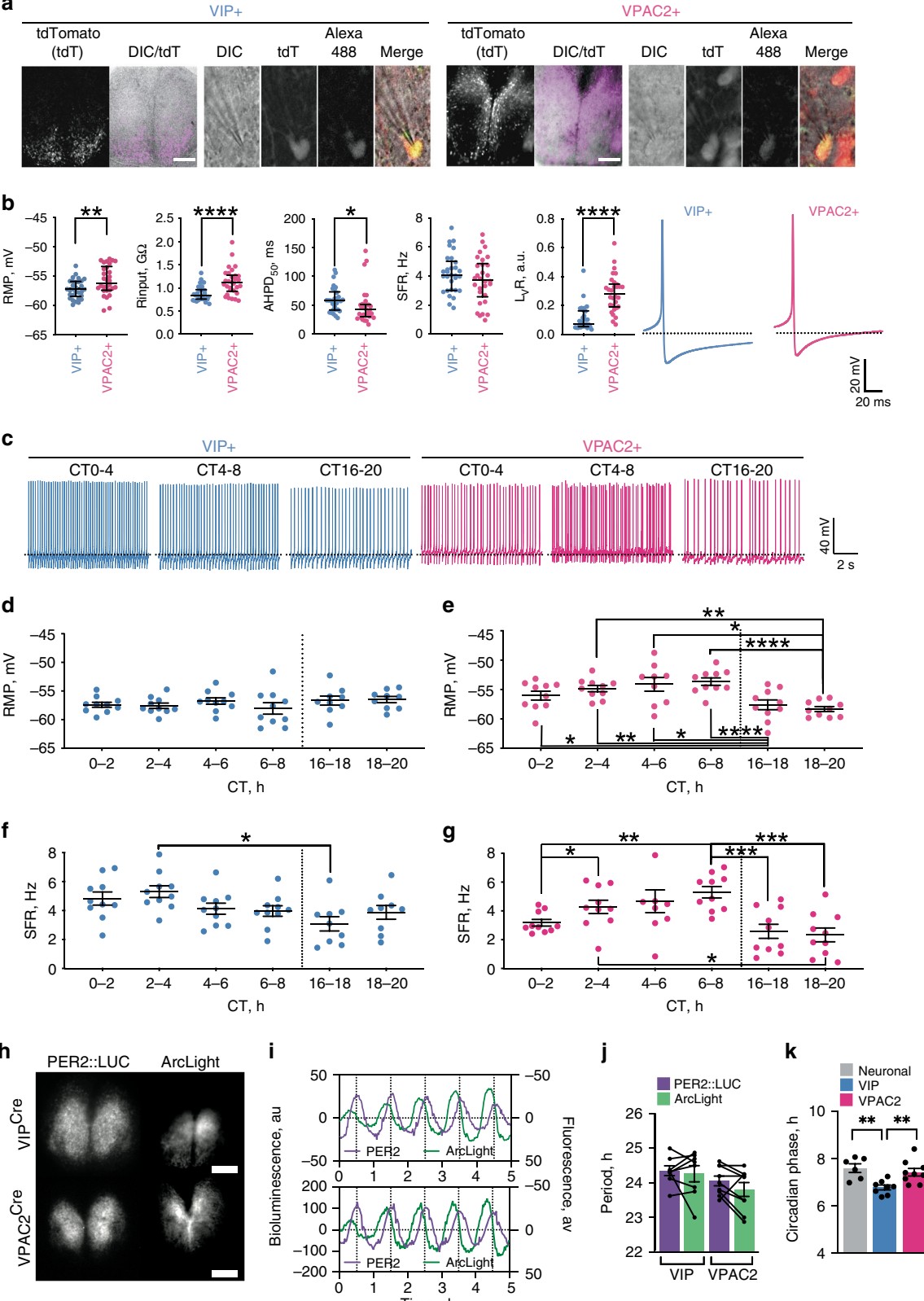

between quiescence (0 Hz) and ~10 Hz, and peaking during mid-circadian day (Supplementary Fig. 5a–c)[24]. We examined the electrophysiological properties of VIP[Cre] and VPAC2[Cre] cells (identified by transduction with AAV encoding flexed TdTomato (Fig. 2a)) by recording in circadian day (CT0-8) or night (CT16-20) to match ensemble SFR peak or nadir, respectively. When

day- and night-time recordings were combined, VIP[TdTom] neurons appeared less electrically excitable than VPAC2[TdTom] neurons, being hyperpolarised (resting membrane potential (RMP)) with a lower input resistance ($R_{Input}$) and a longer after-hyperpolarisation half duration (AHPD$_{50}$) (Fig. 2b). It might be expected, therefore, that VIP[TdTom] neurons fire at lower rates, but

**Fig. 2 Electrophysiological characterisation of VIP[Cre] and VPAC2[Cre] neurons. a** VIP and VPAC2 cells in SCN slices, targetted for recording by Cre-conditionally expressed TdTomato, which was further confirmed by Biocytin-Alexa488 filling of the cells. Scale bar = 250 μm. **b** Summary generic cellular electrophysiological properties from current clamp recordings (pooled by slice) made from VIP (blue) or VPAC2 (pink) cells pooled across all circadian times. From left to right: resting membrane potential (RMP), input resistance ($R_{input}$), after hyperpolarisation half duration (AHPD$_{50}$), spontaneous firing rate (SFR) and firing regularity (L$_V$R) followed by representative individual action potential waveforms. Each point represents the mean value for an individual slice ($n = 30$ slices per genotype, mean of 12 cells per slice), and lines represent median of the slices ±IQR (*$p = 0.045$, **$p = 0.003$, ****$p < 0.0001$). **c** Example current clamp recordings for VIP (blue) and VPAC2 (pink) cells made during day-time (CT0-4, CT4-8) and night-time (CT16-20) (dotted line: −55 mV). **d–g** Phase-aligned recordings of: **d, e** RMP of VIP (**d**) and VPAC2 cells; **f, g** SFR for VIP (**f**) and VPAC2 (**g**); Blue and pink represent VIP and VPAC2 respectively (*$p < 0.05$, **$p < 0.01$, ***$p < 0.001$, ****$p < 0.0001$). **h** Representative micrographs of Per2::Luciferase in register with ArcLight conditionally targetted to VIP (upper) or VPAC2 (lower) cells. Scale bars = 250 μm. **i** Representative detrended Per2::Luciferase (purple) and conditionally targetted inverted ArcLight traces (green) from recordings from VIP[Cre] (upper) and VPAC2[Cre] (lower) slices. **j** Paired period measures of Per2::Luciferase and conditional ArcLight rhythms. **k** Circadian-normalised phase measures of peak depolarisation for pan-neuronal ArcLight (grey), VIP[GCaMP6f] (blue) and VPAC2[GCaMP6f] (pink) (**$p < 0.01$). In all histograms, each point represents the mean value for an individual slice, while the histogram bar/line represents the mean for all of the slices ±SEM. For time-aligned electrophysiology (**d–g**): pooled slice level data for $n = 10$ slices per genotype per time point (except for VIP CT16-18, VIP CT18-20, and VPAC2 CT4-6, where $n = 9$ slices per genotype per time point); for ArcLight imaging (**j, k**): $n = 6$, 8 and 9 for pan-neuronal, VIP[ArcLight] and VPAC2[ArcLight]. Statistics: **b, d–g** linear mixed model followed by: **b** Two-tailed Mann–Whitney $U$ test; **d–g** two-way ANOVA with Tukey's correction for multiple comparisons; **j** paired two-tailed $t$-test; **k** ordinary one-way ANOVA with Holm-Sidak's correction for multiple comparisons. Only significant comparisons ($p < 0.05$) are shown. Source data are provided as a source data file.

overall SFR was not different between cell types (Fig. 2b). Firing regularity, assessed as L$_V$R[25], was, however, different: VIP[TdTom] neurons were highly regular whereas VPAC2[TdTom] discharge was irregular (Fig. 2b).

When compared between circadian day and night, the contrasting regularity of firing in VIP and VPAC2 cells ($p < 0.0001$) neither changed with time, nor was there an interaction ($p = 0.14$) between time and cell type (Supplementary Fig. 5d, e). Regularity of firing is, therefore, not circadian, but an intrinsic property distinguishing VIP and VPAC2 neurons. In contrast, the temporal pattern of RMP differed significantly between VIP and VPAC2 cells (Fig. 2c–e) (ANOVA, time: $p < 0.05$, genotype: $p < 0.001$, interaction: $p < 0.0001$). VPAC2[TdTom] cells were consistently depolarised compared to VIP[TdTom] cells and showed significant hyperpolarisation at night versus day (Fig. 2d, e), whereas VIP[TdTom] cells lacked measurable circadian RMP changes (Fig. 2d). In both populations, SFR was significantly higher in circadian day (CT0-8) versus night (CT16-20) ($p < 0.0001$) (Fig. 2f, g), but with a significant interaction between time and cell type ($p < 0.05$). Whereas SFR of VIP[TdTom] cells peaked at CT0-4 (Fig. 2f), VPAC2[TdTom] cells peaked at CT4-8 indicating that electrical activity in VIP cells is phase advanced relative to VPAC2 cells. To obtain complementary analyses of circuit-level electrophysiological activity, we transduced SCN with the voltage sensor AAV synapsin-ArcLight, expressed either pan-neuronally[19,22] (Supplementary Fig. 5f–h) or conditionally in VIP[Cre] or VPAC2[Cre] cells (Fig. 2h), and phased-registered the fluorescence to simultaneous whole-field Per2::Luciferase recordings. Both VIP[ArcLight] and VPAC2[ArcLight] neurons demonstrated stable circadian rhythms in electrical activity (Fig. 2i, j). Importantly, the peak circadian phase was significantly phase advanced in VIP[ArcLight] cells by ~1 h relative to pan-neuronal and VPAC2[ArcLight] cells (Fig. 2k) (Neuronal: CT 7.6 ± 0.2 h; VIP: CT6.8 ± 0.1 h; VPAC2: CT7.4 ± 0.2 h). Thus, VIP and VPAC2 cells are electrophysiologically distinct and exhibit differentially phased (VIP advanced) circadian electrophysiological rhythms.

**VIP/VPAC2 cellular TTFL-rhythms are differentially phased.** To test whether the differentially phased circadian rhythms of electrical activity and [Ca$^{2+}$]$_i$ in the VIP and VPAC2 subpopulations arose from cell-type-specific TTFL differences, we transduced SCN with a conditional *Cry1* transcriptional reporter (*pCry1*-flexed-Luciferase)[26] (Fig. 3a, Supplementary Fig. 6a). *Cry1*-driven bioluminescence from VIP[Cre] and VPAC2[Cre] SCN slices was strongly circadian

(Supplementary Fig. 6b–f). We then determined TTFL phase by co-registration with pan-neuronal synapsin-GCaMP6f (Fig. 3a–c). Combined bioluminescent/fluorescent CCD recordings confirmed spatially appropriate *Cry1*-Luciferase signals, and phase-aligned bioluminescence curves revealed significant phase differences: VIP[Cry1-Luc] cells peaking ~1.8 h earlier than VPAC2[Cry1-Luc] cells (VIP: CT11.0 ± 0.2 h; VPAC2: CT12.8 ± 0.3 h) (Fig. 3d, e). Interestingly, Rayleigh measurements indicated more phase dispersal in VPAC2[Cry1-Luc] cells than VIP[Cry1-Luc] cells (Fig. 3f, g, Supplementary Fig. 6g, h), echoing their corresponding local [Ca$^{2+}$]$_i$ waves. We therefore quantified the local bioluminescent wave by centre-of-luminescence (CoL) measurements[7,17,19] (Fig. 3h, Supplementary Fig. 6g, h). Similar to the contrasting spatial dynamics of VIP and VPAC2 cells revealed by [Ca$^{2+}$]$_i$ recording, CoL measurements showed cell-type-specific properties to the regional spatio-temporal wave: VIP[Cry1-Luc] bioluminescence exhibited a precisely ordered ventro-dorsal sweep, while VPAC2[Cry1-Luc] bioluminescence spread widely across the SCN shell, with limited spatio-temporal coherence as evidenced by the restricted CoL area (Fig. 3i). Together, these data show that VIP and VPAC2 cells form spatially, neurochemically and electrophysiologically distinct populations, with VIP cells phase advanced to VPAC2 cells by all measures: electrical activity, cytosolic calcium and TTFL. Furthermore, the populations exhibit distinct local stereotypical waves, and individually correspond to separate components of the SCN ensemble wave, suggesting that serial signalling through VIP to VPAC2 cells determines a specific temporal segment of the network, and that the two populations may encode and broadcast different variants of circadian information.

**Control of circuit coherence and phase by VIP/VPAC2 cells.** VIP- and VPAC2-expressing cells have different intrinsic circadian properties. We therefore tested their particular contributions to SCN network function by selective ablation using Cre-conditional diphtheria toxin receptor (DtR-P2A-mCherry). SCN slices transduced with DtR highlighted VIP[Cre] or VPAC2[Cre] cells with spatially appropriate mCherry signals, >99% of which were lost following specific ablation by addition of diphtheria toxin (Dtx) when compared with vehicle controls (Supplementary Fig. 7a–d). Circuit-level circadian function monitored as Per2:: Luciferase rhythms (Fig. 4a, b) was unperturbed in the absence of DtR and/or Dtx. Dtx-mediated ablation of VIP[Cre] or VPAC2[Cre] cells did not affect ensemble period (Supplementary Fig. 7e), but did reduce rhythm amplitude and coherence (Fig. 4a, b,

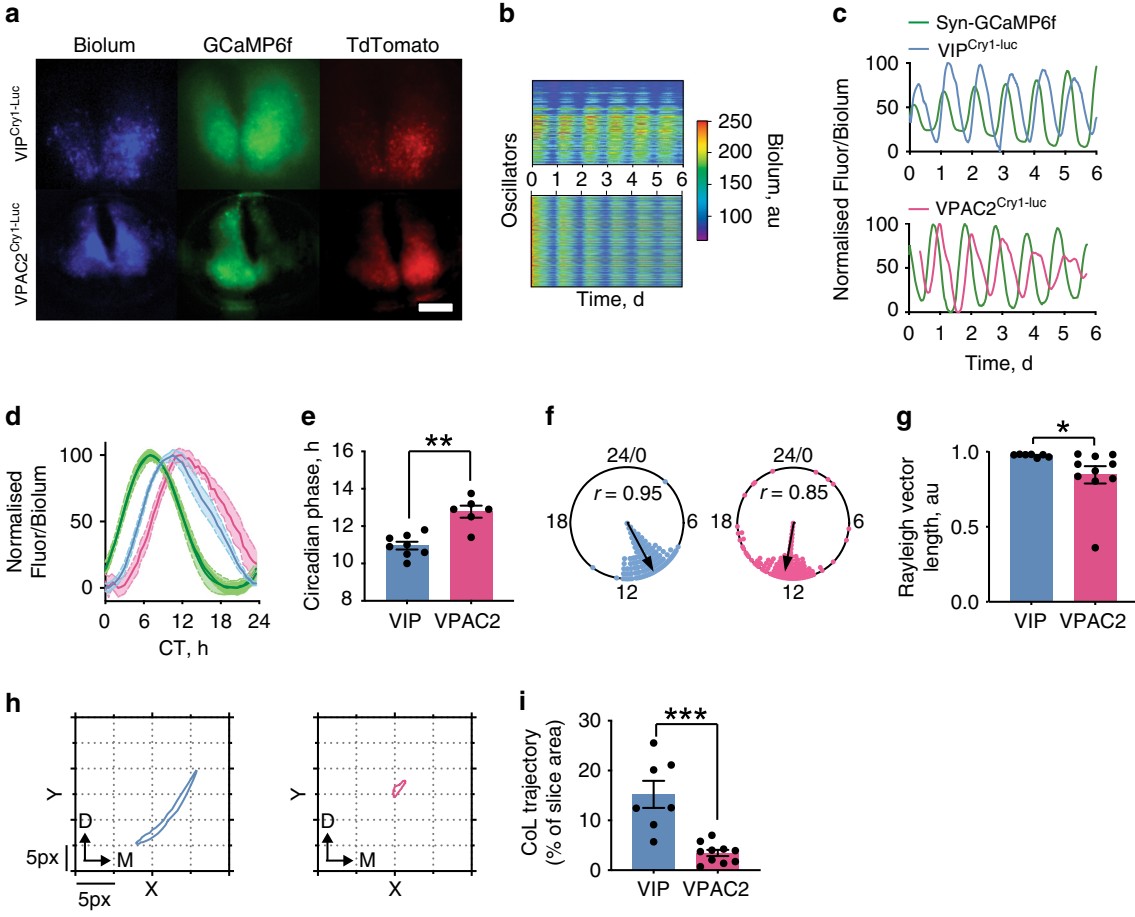

**Fig. 3 Circadian profile of Cry1-luciferase reveals differential TTFL properties of VIP and VPAC2 cells. a** False-coloured micrographs of *pCry1*-Luciferase (blue), pan-neuronal GCaMP6f (green) and conditional TdTomato (red), showing regional bioluminescence and TdTomato signal in VIP[Cre] and VPAC2[Cre] SCN (scale bar = 250 μm) **b** Raster plots showing cellular bioluminescence intensity through time for VIP[Cry1-luc] (upper) and VPAC2[Cry1-luc] (lower) recordings. **c** Representative detrended pan-neuronal GCaMP6f (green) and VIP[Cry1-luc] (blue, upper) and VPAC2[Cry1-luc] (pink, lower) traces. **d** Mean aligned and normalised single cycles of neuronal-GCaMPf (two overlaid green traces) versus corresponding VIP[Cry1-luc] (blue) and VPAC2[Cry1-luc] (pink). Lines and shading are mean ± SEM, *n* = 8 VIP[Cry1-luc] and 6 VPAC2[Cry1-luc]. **e** Circadian-normalised phase measures for VIP[Cry1-luc] (blue, *n* = 8) and VPAC2[Cry1-luc] (pink, *n* = 6) (**\*\**p* = 0.003). **f** Representative Rayleigh plots for VIP[Cry1-luc] and VPAC2[Cry1-luc] oscillators showing the relatively broader coupling in VPAC2[Cry1-luc] recordings. **g** Mean Rayleigh vector lengths for VIP[Cry1-luc] (*n* = 7) and VPAC2[Cry1-luc] (*n* = 10) (\**p* = 0.041). **h** Representative centre of luminescence (CoL) trajectories showing the direction and magnitude of local bioluminescent excursions for VIP[Cry1-luc] (blue, left) and VPAC2[Cry1-luc] (pink, right). D and M arrows indicate dorsal and medial respectively. **i** Summary CoL trajectories expressed as a percentage of slice area covered for VIP[Cry1-luc] (blue, *n* = 7) and VPAC2[Cry1-luc] (pink, *n* = 10) (\*\*\**p* = 0.0004). For all histogram plots: individual points represent individual slices; histogram bars: mean ± SEM. For Rayleigh plots: individual points represent individual oscillator circular phases; arrows: mean circular phase for the slice. Statistics: **e** unpaired two-tailed *t*-test; **g, i** unpaired two-tailed Mann–Whitney *U* test. Only significant comparisons (*p* < 0.05) are shown. Source data are provided as a source data file.

Supplementary Fig. 7f, g). CCD recordings to assess cellular and network function revealed that VIP[Cre] cell ablation restricted the most remaining bioluminescence dorsally, while VPAC2[Cre] ablation restricted remaining bioluminescence to the central and core SCN (Fig. 4c, d). Furthermore, raster- and Rayleigh-plots of VIP-ablated SCN revealed progressively desynchronised cellular oscillations with reduced amplitude, and divergence of individual cellular periods well beyond those of the intact slice. In contrast, VPAC2-ablated SCN showed significantly less phase and period divergence (Fig. 4e–g, Supplementary Fig. 7h–j). Thus, ablation of VIP cells dramatically reduced amplitude and circadian precision, increasing cellular desynchrony and period divergence, while ablation of VPAC2 cells resulted in a largely synchronous SCN with limited phase and period dispersal. VIP and VPAC2 cells therefore contribute differentially to circuit-level functions: loss of VIP cells compromises SCN synchrony more than does loss of VPAC2 cells.

To explore further the ability of VIP and VPAC2 cells to differentially affect the SCN circuit, we implemented optogenetic control of neuronal firing in SCN slices using AAV-Channelrhodopsin-2::EYFP (ChR2::EYFP) (Fig. 5a, Supplementary Fig. 8a). We then tested the ability of direct electrical activation of VIP[Cre] or VPAC2[Cre] cells to control the phase of ensemble Per2:: Luciferase rhythms by constructing phase response curves (PRCs) to blue (470 nm) or control red (625 nm) light stimulation (1 h at 10 Hz). Optogenetic activation of VIP[ChR2::EYFP] cells caused large phase-delays in early circadian night and small, non-significant phase-advances in late circadian night (Fig. 5b, c) (red light had no significant effect)[13]. Optogenetic activation VPAC2[ChR2::EYFP] cells, however, did not initiate significant phase-shifts (Fig. 5d, e). Direct comparison of VIP[ChR2::EYFP] and VPAC2[ChR2::EYFP] PRCs therefore revealed highly significant time effects and interaction between time and cell-type (both *p* < 0.01) (Fig. 5f). Thus, specific optogenetic activation of VIP cells was sufficient to set SCN

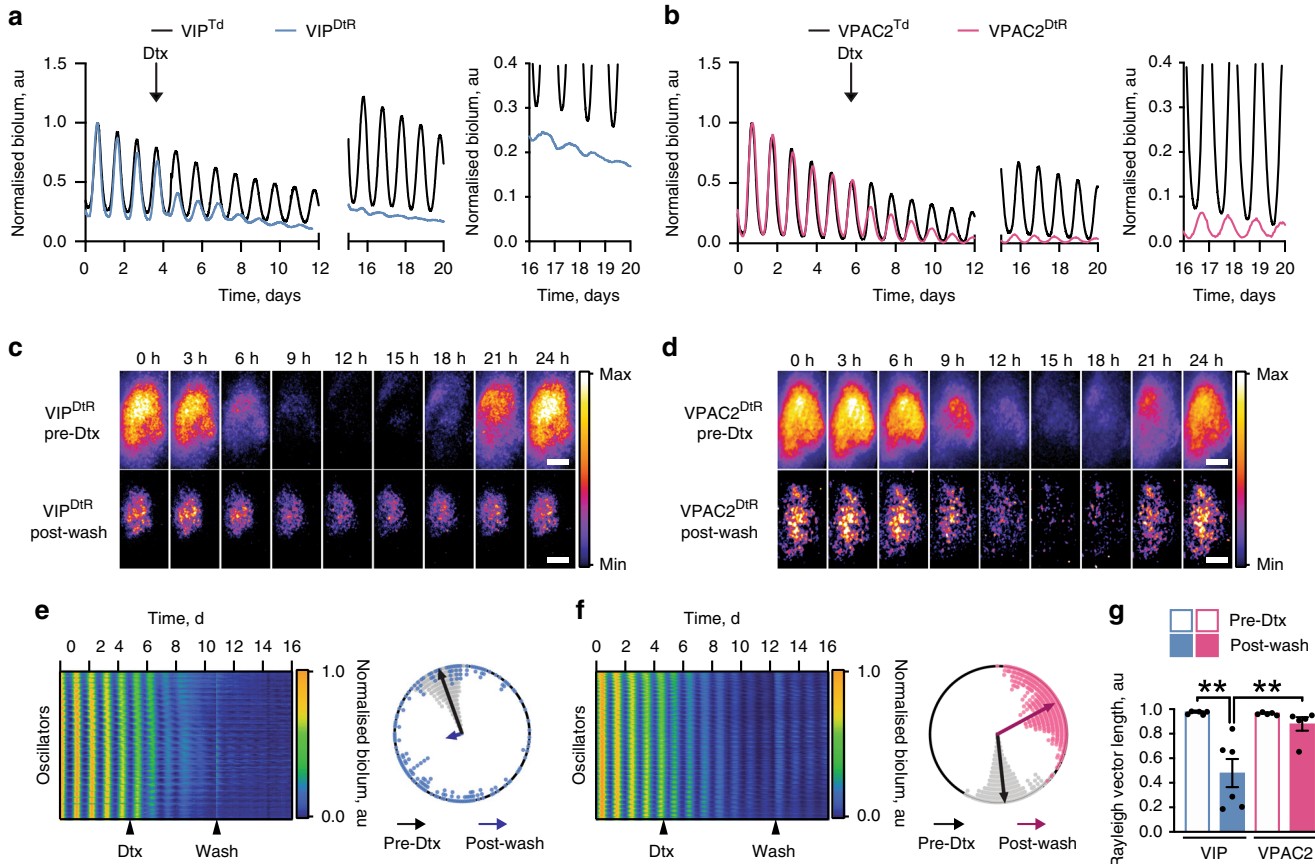

**Fig. 4 Differential contributions of VIP cells and VPAC2 cells to SCN circuit-level function revealed by their specific toxin-mediated ablation.**
**a, b** Representative normalised Per2::Luciferase traces for (**a**) VIP^{Cre} SCN and (**b**) VPAC2^{Cre} SCN conditionally expressing either TdTomato (Black) or the diphtheria toxin receptor (DtR) (VIP^{Cre}: blue, VPAC2^{Cre}: pink). The traces show ongoing ensemble oscillation before, during and after wash-out of diphtheria toxin (Dtx) addition, and the insets to the right show expanded views of the oscillations post-washout. **c, d** Bioluminescent micrographs of a single cycle of bioluminescence for (**c**) VIP^{DtR} and (**d**) VPAC2^{DtR} SCN before addition of Dtx (upper) and post-washout (lower). Scale bars = 100 μm. **e, f** Raster plots showing cellular bioluminescence intensity through time for (**e**) VIP^{DtR} and (**f**) VPAC2^{DtR} SCN recordings conditionally expressing DtR before, during and after addition of Dtx. Alongside are representative Rayleigh plots of the corresponding slices showing phase-coherence before addition of Dtx (black) and post-washout (VIP^{Cre}: blue; VPAC2^{Cre}: pink). Individual points represent individual oscillator circular phases; arrows represent mean circular phase for the slice and condition. **g** Summary Rayleigh vector lengths before Dtx treatment (hollow bars) and post-washout (solid bars) for VIP^{Cre} (blue, n = 6) and VPAC2^{Cre} (pink, n = 5) SCN conditionally expressing DtR (**p < 0.01). Individual points represent individual slice values; bars represent mean ± SEM. Statistics: **g** repeated-measures two-way ANOVA with Sidak's correction for multiple comparisons. Only significant comparisons (p < 0.05) are shown. Source data are provided as a source data file.

ensemble phase, whereas electrical activation of their VPAC2 target cells was not.

If resetting by VIP cells cannot be explained solely by increased firing of their VPAC2 targets, molecular and/or metabolic actions of VIP additional to electrical activation likely effect resetting of the TTFLs of VPAC2 cells. The most direct way would be VIP-dependent acute induction of *Per1* and *Per2*, mediated by cAMP/calcium responsive elements (CREs)[15]. To explore this, we monitored acute changes in Per2::Luciferase bioluminescence in arrhythmic Cry1,2-null (Cry-null) SCN, which provided a stable baseline to observe TTFL responses induced by activation of VIP cells. In Cry-null SCN, acute pan-neuronal optogenetic activation elevated bioluminescence, peaking after ~3–4 h and returning to baseline after ~10-11 h (Fig. 5g). Stimulation solely of VIP^{ChR2::EYFP} cells in Cry-null SCN resulted in a comparable bioluminescent induction (Fig. 5h). Furthermore, using ChR2::EYFP to delineate the localisation of VIP cells and cross-correlating maps of bioluminescent change with EYFP intensity revealed that optogenetic activation induced Per2 expression both inside and outside the territory of VIP cells, with the same magnitude, kinetics and distribution of responses (both p > 0.2) (Fig. 5i–l, Supplementary

Fig. 8b). Thus, output from VIP cells activates and recruits pan-SCN cellular TTFLs, with serial activation along the VIP/VPAC2 cellular axis re-setting ensemble phase.

**VIP/VPAC2 cells control period and ab initio oscillation.** The VIP/VPAC2 cellular axis contributes to SCN spatio-temporal dynamics, maintains cellular synchrony and controls ensemble phase. The final emergent properties are circadian oscillation and ensemble period. The ability of VIP and/or VPAC2 cells to determine ensemble period was tested by genetic complementation of short-period Cry1-null SCN using conditional AAV expression of Cry1 (*pCry1*-DIO-Cry1::EGFP) to lengthen their cell-autonomous period[26–28]. If VIP^{Cre} or VPAC2^{Cre} cells have pace-setting properties, then following transduction the ensemble period should lengthen from ~22 h to ~24 h to match their newly established, Cry1-dependent periods[27]. Expression of Cry1 in VIP^{Cre} cells, VPAC2^{Cre} cells, or in both populations simultaneously was confirmed by appropriate nuclear EGFP signal, with cell-type specificity confirmed by co-localisation with Cre-dependent mCherry expression (*pSyn*-DIO-mCherry). The transduction efficiency for both AAVs was over 90% of targeted

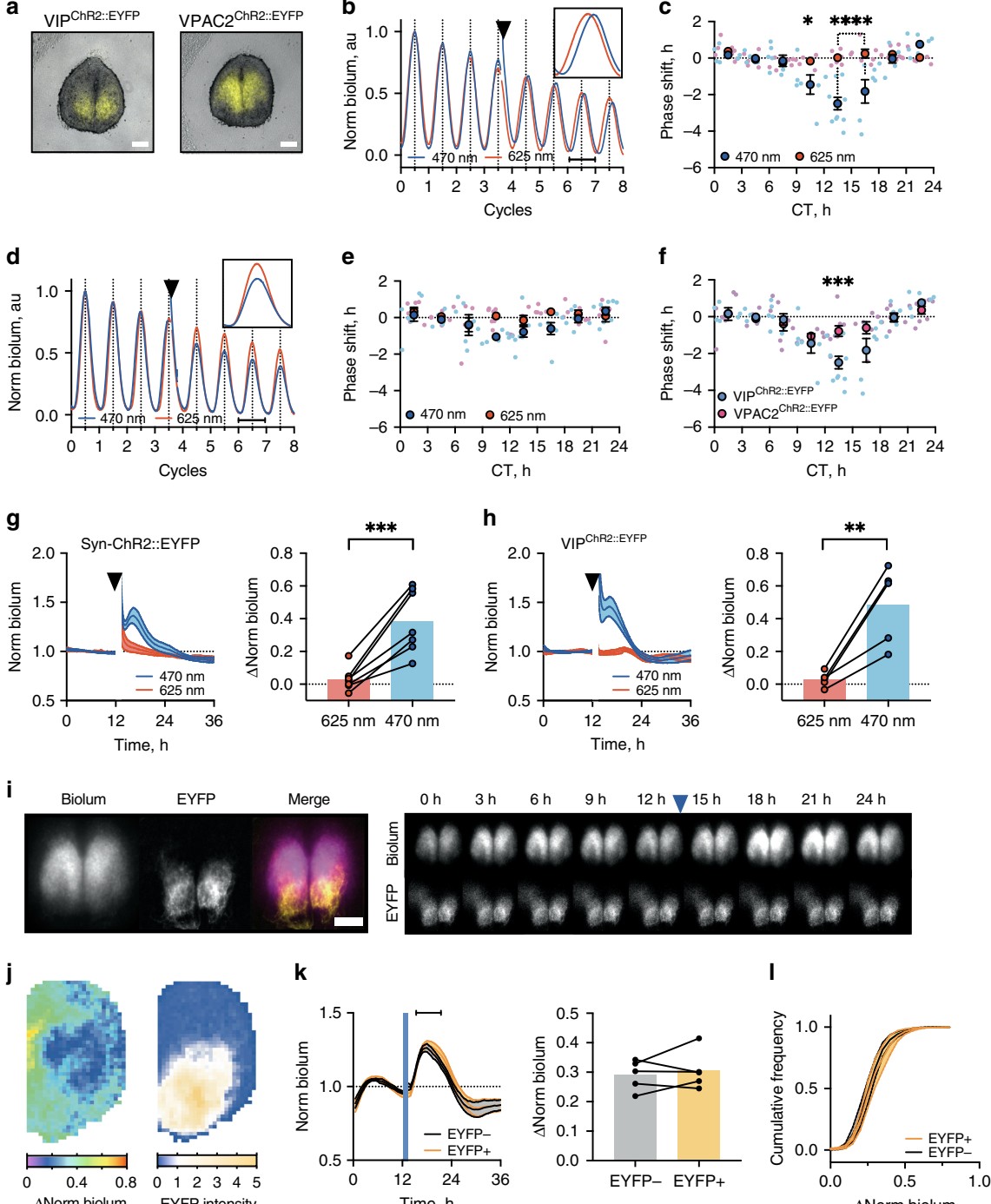

cells (Supplementary Fig. 9). Expression of Cry1 in VIP[Cre] cells caused a small, but significant, period lengthening by ~1 h (Fig. 6a, d) whilst expression in VPAC2[Cre] cells was ineffective (Fig. 6b, d). In contrast, expression in both populations progressively lengthened period, stabilising at ~24.5 h (Fig. 6c, d) comparable to the pacesetting effect of untargetted, pan-neuronal Cry1::EGFP expression[24]. Thus, although VIP cells have a small influence and VPAC2 cells none, VIP and VPAC2 cells together can determine SCN ensemble period.

A further test of pacemaking is whether cells can initiate and sustain rhythmicity in otherwise circadian-incompetent SCN: are VIP and/or VPAC2 cells able to control the network if they are the only cell-autonomous oscillators? To address this, Cre-conditional Cry1 was expressed in arrhythmic Cry-null SCN to activate cell-autonomous TTFLs in VIP[Cre] cells, VPAC2[Cre] cells or both. Complementation of the cell-autonomous TTFLs of individual VIP[Cre] or VPAC2[Cre] populations could not initiate coherent rhythms (Fig. 7a). Activation of both groups, however, rapidly (~4 days) initiated rhythmicity, and with a long period comparable to rhythms initiated by pan-neuronal (pSyn-Cre-mediated) expression of Cry1 (Fig. 7a, b). The oscillation was of high quality, assessed by goodness-of-fit (GOF), again approaching that of SCN subject to pan-neuronal expression of Cry1[27] (Fig. 7c). Importantly, the oscillation is not simply due to mass action by recruitment of a larger population of cells: transduction of comparable numbers of VPAC2 cells alone did not improve

**Fig. 5 Targetted optogenetic activation of VIP cells sets ensemble phase, whereas activation of VPAC2 cells does not. a** Representative merged fluorescence/brightfield micrographs of VIP$^{ChR2::EYFP}$ (left) and VPAC2$^{ChR2::EYFP}$ (right). **b** Representative 470 nm (blue) or 625 nm (red) stimulated normalised Per2::Luciferase trace from VIP$^{ChR2::EYFP}$ SCN. Inset: phase-delay third cycle post-stimulation. **c** Phase response curve (PRC) for VIP$^{ChR2::EYFP}$ SCN stimulated at 470 nm (blue) or 625 nm (red/pink) (*$p = 0.032$, ****$p < 0.0001$). **d, e** As in (**b, c**), but for VPAC2$^{ChR2::EYFP}$ SCN. **f** Combined PRCs for 470 nm-stimulated VIP$^{ChR2::EYFP}$ (blue) and VPAC2$^{ChR2::EYFP}$ (pink) SCN (***$p = 0.0002$). **g, h** Normalised bioluminescence traces for Cry-null SCN expressing (**g**) pan-neuronal ChR2::EYFP or (**h**) VIP$^{ChR2::EYFP}$ stimulated at 470 nm (blue) or 625 nm (red) alongside paired summary data (right) (**$p = 0.008$, ***$p = 0.0009$). **i** Micrographs showing VIP$^{ChR2::EYFP}$ Cry-null SCN (Per2::Luciferase, ChR2::EYFP and merge) (left) alongside Per2::Luciferase time-series (upper) and ChR2::EYFP (lower) before and after optogenetic stimulation. **j** Example spatial bioluminescence change (left) and VIP$^{Cre}$-dependent EYFP intensity (right) in VIP$^{ChR2::EYFP}$ SCN slice. **k** Averaged bioluminescence traces within (EYFP intensity > 1, orange) or outside (EYFP intensity < 1, blue) EYFP regions (left) in VIP$^{ChR2::EYFP}$ SCN, with paired summaries (right) measured over interval indicated by horizontal line. **l** Cumulative distributions of bioluminescent changes within (orange) or outside (grey) EYFP regions in VIP$^{ChR2::EYFP}$ SCN. Scale bars = 250 μm. Arrowheads: stimulation. PRCs: small points represent individual slices; large points represent mean ± SEM. Line plots: lines and shading represent mean ± SEM. Histograms: paired points represent individual slices; bars represent mean ± SEM. PRCs: $n \geq 5$ for each phase, explicit $n$ values are given in Supplementary Table 1; Cry-null recordings: PMT: neuronal ChR2::EYFP $n = 7$; VIP$^{ChR2::EYFP}$ $n = 6$ SCN; Imaging: $n = 5$ SCN. Statistics: **c**, b, f two-way ANOVA; **g, h, k**, paired two-tailed $t$-test; **l** repeated-measures two-way ANOVA. Only significant comparisons ($p < 0.05$) are shown. Source data are provided as a source data file.

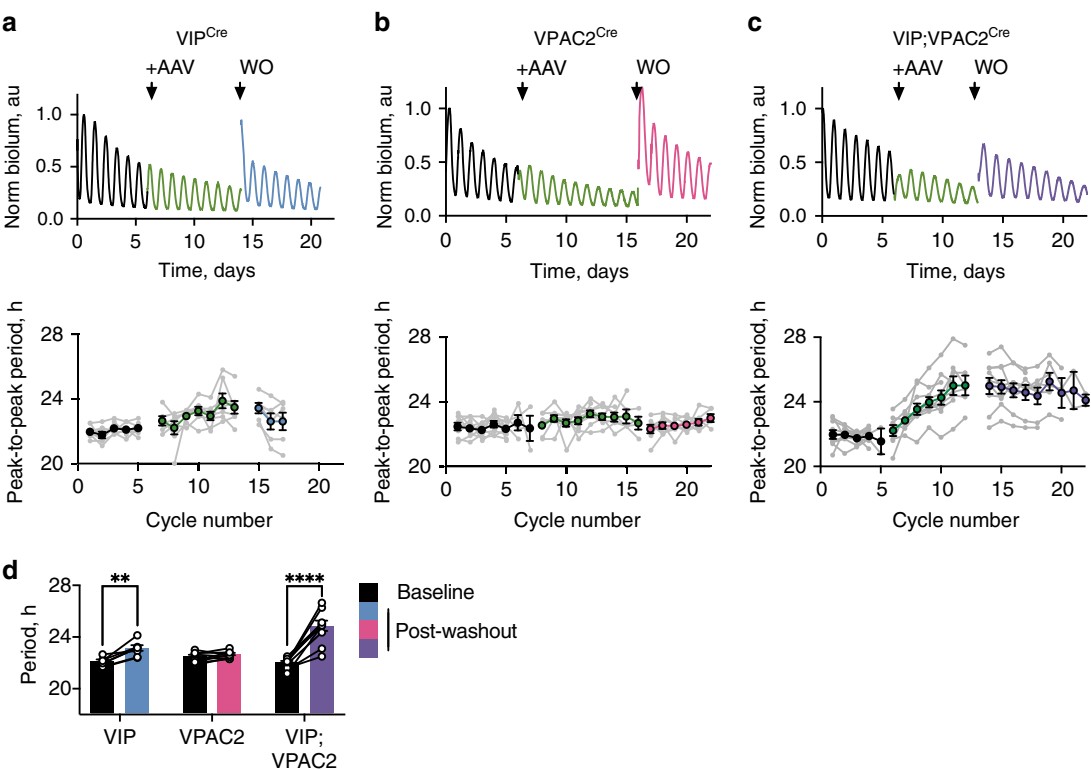

**Fig. 6 Conditional Cry1 expression in the VIP-VPAC2 cellular axis of *Cry1*-null SCN is sufficient to determine ensemble period of the SCN. a, c** Upper: representative raw Per2::Luciferase PMT traces of Cry1-null SCN before (black) and during (green) initiation of conditional Cry1-expression, and following AAV washout (WO) (coloured by genotype: VIP$^{Cre}$ (blue), VPAC2$^{Cre}$ (pink) and VIP$^{Cre}$;VPAC2$^{Cre}$ (purple)). Lower: corresponding peak-to-peak period plots (mean ± SEM) for VIP$^{Cre}$, VPAC2$^{Cre}$ and VIP$^{Cre}$;VPAC2$^{Cre}$ SCN. Arrow indicates addition of conditional Cry1 AAV. For peak-to-peak period plots, joined grey points represent individual slices; coloured points represent mean ± SEM with colours denoting treatment interval in traces as above. **d** Summary period data for: pre-AAV-Cry1 baseline (black) and post-AAV-Cry1 washout for VIP$^{Cre}$ (blue), VPAC2$^{Cre}$ (pink) and VIP$^{Cre}$;VPAC2$^{Cre}$ (purple) SCN (**$p = 0.007$, ****$p < 0.0001$). Histograms: paired points represent individual slices; bars represent mean ± SEM ($n = 7$, 9 and 10 for VIP$^{Cre}$, VPAC2$^{Cre}$ and VIP$^{Cre}$;VPAC2$^{Cre}$). Statistics: **d** repeated-measures two-way ANOVA with Sidak's correction for multiple comparisons. Only significant comparisons ($p < 0.05$) are shown. Source data are provided as a source data file.

rhythm quality (Fig. 7d). Rather, it was due to the specific recruitment of both cellular constituents of the VIP neuropeptidergic axis.

Initiation of ensemble rhythmicity demonstrated the clear pacemaking property of the VIP/VPAC2 cellular axis. We then used CCD recordings to test whether it also directed further network features, including synchrony and spatio-temporal dynamics. CoL measurements revealed the absence of a spatiotemporal wave in Cry1-transduced, single-Cre SCN (Fig. 7e). In contrast, joint activation of cell-autonomous TTFLs in VIP$^{Cre}$

and VPAC2$^{Cre}$ cells initiated strong Per2::Luciferase bioluminescent waves, approaching wild-type trajectories and accompanied by tight phase coupling of oscillators across the network (Fig. 7e, f, Supplementary Fig. 10). Furthermore, phase maps of Per2:: Luciferase bioluminescence in Cry1-initiated slices recapitulated many of the spatiotemporal dynamics of the wild-type slice, including region-specific phasing (Fig. 1), with the induced activity starting dorsomedially and progressing to the core before moving dorsally, medially and laterally into the phase-delayed shell (Fig. 7g, Supplementary Fig. 10).

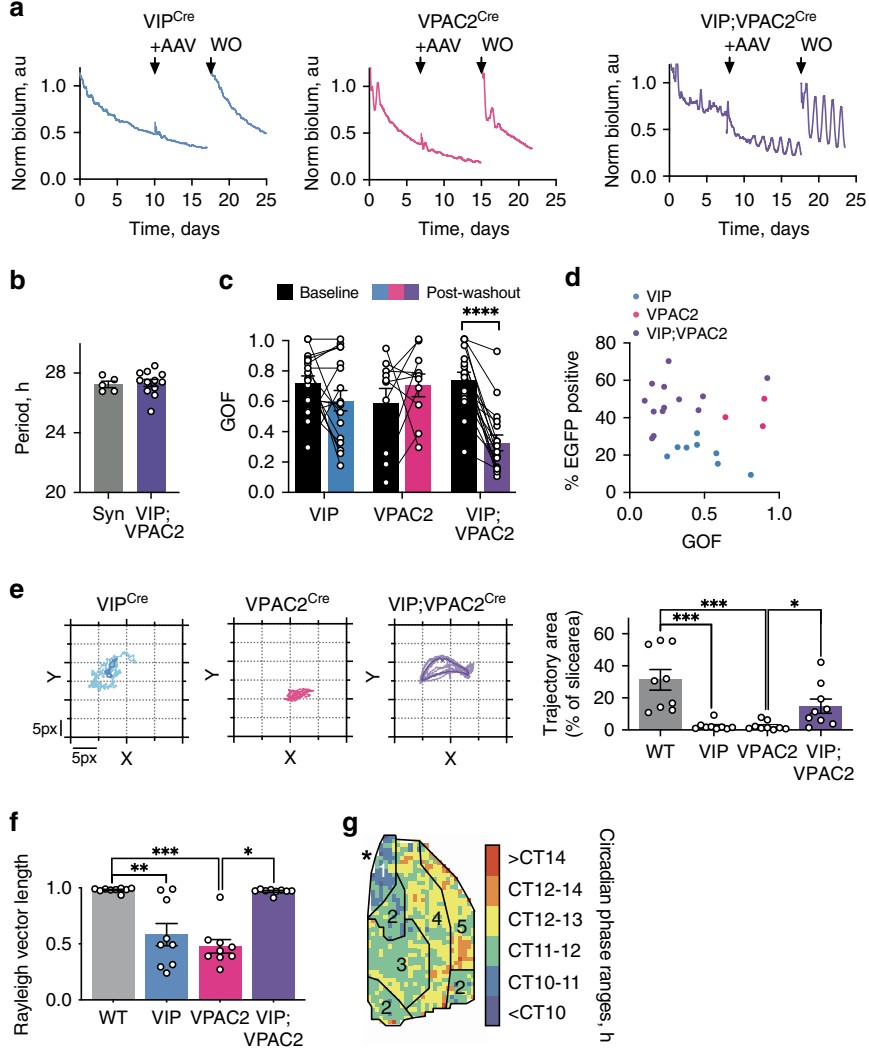

**Fig. 7 Conditional Cry1 expression in the VIP-VPAC2 cellular axis of Cry-null SCN is sufficient to initiate and sustain circadian rhythmicity. a** Representative Per2::Luciferase PMT traces of Cry-null SCN, before and during initiation of conditional Cry1-expression, and after AAV washout (WO) in VIP[Cre] (blue, left), VPAC2[Cre] (pink, middle) and VIP[Cre];VPAC2[Cre] slices (purple, right). Arrows indicate AAV addition and washout. **b** Summary periods for pan-neuronal (Syn, grey, $n = 5$) or VIP[Cre];VPAC2[Cre] (purple, $n = 12$) Cry1 expression. **c** Paired goodness of fit (GOF) measures for baseline (black) and post-washout (coloured) for VIP[Cre] (blue, $n = 18$), VPAC2[Cre] (pink, $n = 10$) and VIP[Cre];VPAC2[Cre] (purple, $n = 17$) SCN ($p < 0.0001$). **d** Percentage of Cry1:: EGFP positive SCN cells versus the GOF for VIP[Cre] (blue, $n = 8$), VPAC2[Cre] (pink, $n = 3$) and VIP[Cre];VPAC2[Cre] (purple, $n = 13$) conditional Cry1-expressing SCN. **e** Representative CoL trajectories showing relative spatial organisation of VIP[Cre] (blue, $n = 9$), VPAC2[Cre] (pink, $n = 9$) and VIP[Cre];VPAC2[Cre] (purple, $n = 9$) in Cry-null SCN conditionally expressing Cry1 (left) alongside summary trajectories (right) including wild type (grey, $n = 9$) (*$p = 0.047$, ***$p < 0.001$). Lighter lines indicate individual cycles, darker lines indicate the mean of the individual cycles. **f** Summary Rayleigh vector length for wild-type SCN (grey, $n = 9$), or VIP[Cre] (blue, $n = 9$), VPAC2[Cre] (pink, $n = 9$) and VIP[Cre];VPAC2[Cre] (purple, $n = 9$) Cry-null SCN conditionally expressing Cry1 (*$p = 0.027$, **0.004, ***$p = 0.0002$). **g** Representative categorical, annotated phase map showing spatial phase distribution in VIP[Cre];VPAC2[Cre] Cry1-initiated SCN, asterisk indicates third ventricle. In all plots: Individual points (paired/unpaired) represent individual slices; bars represent mean ± SEM. Statistics: **b** unpaired two-tailed t-test, **c** repeated-measures two-way ANOVA; **e, f** Ordinary one-way ANOVA. Only significant comparisons ($p < 0.05$) are shown. Source data are provided as a source data file.

Joint recruitment of the TTFLs of the VIP and VPAC2 cells alone set the pace and initiated the ex vivo SCN ensemble rhythm. We therefore asked if these populations are sufficient to transmit rhythmic behaviour to the animal. Male and female Cry-null mice, expressing Cre in VIP and VPAC2 cells, were arrhythmic under constant darkness (DD), and no period could be assigned (Fig. 8a). They then received bilateral stereotaxic injections of the *pCry1*-DIO-Cry1::EGFP AAV, and post hoc histology showed that of the eight mice receiving stereotaxic injections, six were successfully targeted, with Cry1::EGFP expression that spanned across the ventral core and dorsal shell of the SCN (Fig. 8b). The remaining two mice exhibited limited

Cry1::EGFP expression in the very ventral SCN alone (Supplementary Fig. 11a). Following administration of the Cry1-expressing AAV and return to DD, the targeted mice, with one exception, initiated a significant circadian behavioural rhythm, with a long period typical of a Cry1-dependent SCN (27.0 ± 0.1 h, $n = 5$). This was associated with a significant increase in the periodogram amplitude and non-parametrically determined relative amplitude of behavioural rhythms (Fig. 8c, d). Circadian behaviour was not initiated in the two mis-targeted mice. Of the five of six targeted mice in which circadian behaviour was initiated, their long period was directly comparable to that of Cry-null SCN slices transduced with AAV-Cry1::EGFP (Figs. 7b, 8e).

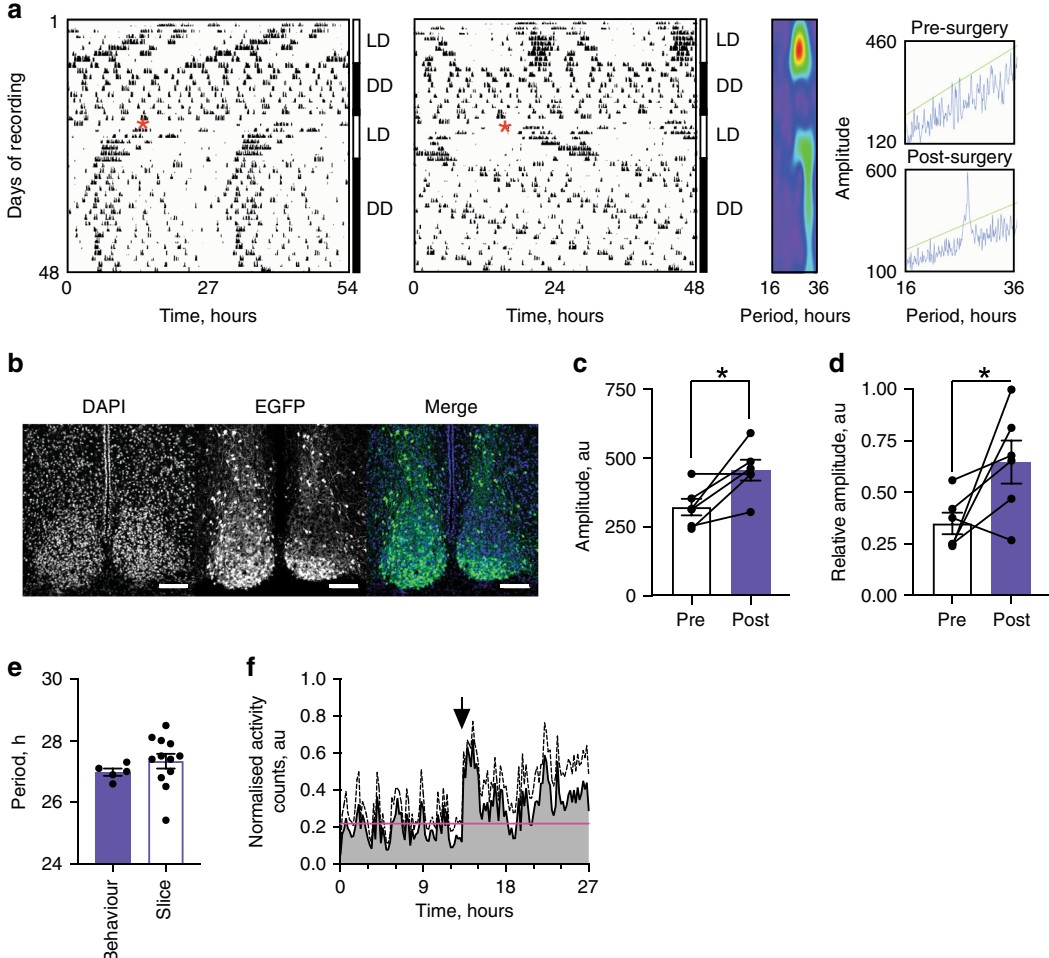

**Fig. 8 Conditional Cry1 expression in the VIP-VPAC2 cellular axis is sufficient to initiate and sustain circadian behaviour. a** Representative double plotted wheel-running traces on a 27-h time base (left) and a 24 h time base (right) showing the behaviour of a VIP;VPAC2$^{Cre}$ Cry-null mouse pre- and post-surgery (indicated by a red asterisk). Wavelet analyses are shown to the right of the plot. **b** Representative confocal images from the SCN of the mouse in **a** showing Cry1::EGFP distribution across the core and shell of the SCN, showing DAPI (left), EGFP (middle) and a false colour merged image (right). Scale bar is 100 μm. **c** Summary behavioural amplitude for VIP;VPAC2$^{Cre}$ Cry-null mice pre- and post-surgery ($n = 6$ mice) (*$p = 0.011$). **d** Summary behavioural relative amplitude for VIP;VPAC2$^{Cre}$ Cry-null mice pre- and post-surgery ($n = 6$ mice) (*$p = 0.034$). **e** Summary free-running periods for VIP;VPAC2$^{Cre}$ Cry-null post-surgery plotted alongside the periods of initiated VIP;VPAC2$^{Cre}$ Cry-null ex vivo SCN (replotted from Fig. 7b) ($n = 5$ mice, 12 SCN). **f** Average normalised activity profiles for the last 7 days of recording post-surgery shown as mean + SEM. The activity of each mouse is aligned at activity onset (indicated by the black arrow), and the mean activity from the pre-surgery condition (subjected to the same relative alignment, shown in Supplementary Fig. 11b) is plotted as a horizontal line (pink) ($n = 5$ mice). In all histograms individual points (paired/unpaired) represent individual mice/SCN slices; bars represent mean ± SEM. Statistics: **c**, **d** paired one-tailed *t*-test, **e** unpaired two-tailed *t*-test. Only significant comparisons ($p < 0.05$) are shown. Source data are provided as a source data file.

In addition, when the behaviour of these mice was plotted as a normalised activity profile and aligned to activity onset across animals, there was a clear division of activity between subjective day and subjective night (% total activity: Day $35.9 \pm 0.04$ vs Night $64.1 \pm 0.04$) (Fig. 8f) compared to pre-surgery where arrhythmic animals lacked this definitive structure (% total activity: Day $53.6 \pm 0.03$ vs Night $46.4 \pm 0.03$) (Supplementary Fig. 11b). Finally, the breeding cohorts generated two Cry1-null, Cry2-proficient mice which had characteristic short period behavioural rhythms. Following successful targeting of their SCN by AAV-Cry1::EGFP (Supplementary Fig. 12), their free-running period lengthened in a manner comparable to that seen in ex vivo Cry1-null SCN slices. These experiments therefore reveal that not only are the VIP and VPAC2 cells, together, driving components of the SCN network ex vivo, but also they are sufficient, in the absence of any other cellular clocks, to direct the circadian behaviour of the animal.

This study has provided a comprehensive characterisation of the role of cells within the VIP/VPAC2 neuropeptidergic axis in determining the emergent properties of the SCN circuit. VIP and VPAC2 cells are neurochemically and electrophysiologically distinct with differentially phased population rhythms that contribute to SCN spatio-temporal dynamics (Figs. 1–4, 9). Cell-specific ablation or activation of VIP or VPAC2 revealed that signals from the VIP core are much more important in maintaining network function and ensemble phase than are signals derived from VPAC2 cells (Figs. 5–6, 9). Finally, although VIP cells exert weak pacemaking control, full period control and de novo initiation of stable high-amplitude rhythms require both compartments of the VIP-ergic cellular axis: showing that the TTFL of VPAC2 cells is necessary for circuit function (Figs. 7–9). Finally, these effects seen ex vivo in the slice can be recapitulated in animals to initiate circadian behaviour: when they contain the only cell-autonomous clocks in the animal, VIP and VPAC2 cells

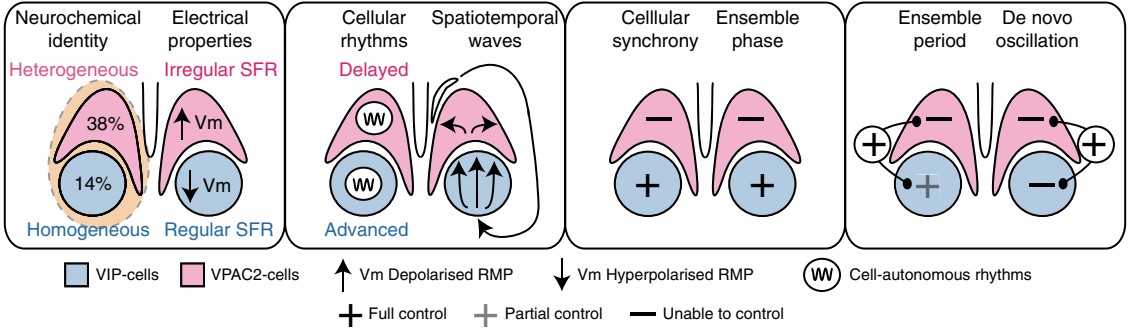

**Fig. 9 The VIP-VPAC2 neuropeptidergic axis as a pacemaking hub within the suprachiasmatic nucleus circadian circuit.** Summary diagram showing the cellular properties of the VIP-VPAC2 neuropeptidergic axis and their control of SCN emergent properties. VIP and VPAC2 neurons are neurochemically, electrophysiologically and spatially distinct SCN subpopulations. VIP neurons are phase advanced relative to VPAC2 neurons with a tight spatiotemporal organisation exhibiting dorsal to ventral waves, while the VPAC2 neurons are phase delayed with a looser medial and lateral spatiotemporal distribution. VIP cells can control cellular synchrony, ensemble phase and weakly contribute to ensemble period while VPAC2 cells cannot definitively control any of these emergent properties. However, both populations together are required to exert pacemaking control and initiation of de novo rhythmicity.

together can direct circadian behaviour (Fig. 8). Thus, emergent, ensemble properties of the SCN network are conferred by circadian signalling via the VIP/VPAC2 axis, which is a cellular pacemaking hub within the SCN circadian circuit.

## Discussion

Recent single-cell SCN transcriptomic analysis has identified putative topological elements involving peptides (AVP, Prok2, PACAP, and VIP) and their cognate receptors[18] that mediate paracrine and/or autocrine signalling[29,30]. Through direct experimental manipulation, we now show that one of these molecularly defined elements, the VIP/VPAC2 cellular axis, consists of serially active topological components of the SCN circuit and determines circuit-level emergent properties of the network.

VIP and VPAC2 cells represent diverse neuronal populations[31,32] and they have distinct electrical signatures that recapitulate those reported for VIP and generic non-VIP cells[13,33], and the difference between VIP and VPAC2 neurons in their intrinsic cellular property of firing regularity is circadian invariant[13]. In both VIP and VPAC2 cells, however, RMP and SFR are rhythmic, peaking during the middle of circadian day, but VIP cells are more hyperpolarised with a lower input resistance and longer AHP than VPAC2 cells. Thus, these two cell populations, when recorded within an intact circuit, have distinctive intrinsic electrophysiological signatures. Furthermore, although the VIP and VPAC2 cells are rhythmically active[13,33], they occupy different phases within the SCN cycle, as inferred from co-registration with phase maps of Per2::Luciferase and demonstrated directly by electrophysiological recordings. VIP cells are phase advanced compared to VPAC2 cells, an effect also seen for RMP, $[Ca^{2+}]_i$, and the TTFL cycle ($pCry1$-Luciferase report). Thus, VIP and VPAC2 cells are serial components within a more complete ensemble phase wave. Moreover, within each cell population there are local waves, most evident as a dorso-ventral trajectory in tightly synchronised VIP cells, whereas VPAC2 cells are less synchronised with a weaker local wave that possibly reflects heterogeneity within the VPAC2 population (e.g. AVP and non-AVP cells). Further analysis of cell-type specific circadian activity, focusing on neuropeptidergically defined topological elements[18] could reveal how the complete ensemble phase wave is generated by serial local activations. This would then inform how circadian information is generated within, and propagated across, the multi-layered SCN cellular network[2].

Previous reports have established that calcium rhythms across the SCN network are phase dispersed[22,23,34], but inconsistent

with our findings, voltage across the network has been reported as simultaneous[22]. These results were based on ArcLight recordings where VIP-cells, AVP-cells or coarse ventral/dorsal ROIs were used as proxies for core and shell SCN, respectively[22]. Here, we utilised two functionally relevant components (VIP and VPAC2) to delineate core and shell to reveal differential phasing between these two compartments, an observation that is consistent with reported phase staggering of the electrical activities of individual neurons across the circadian cycle[12]. The difference we observed is of a smaller magnitude than that reported by calcium, but follows the same pattern: VIP cells are phase-advanced relative to VPAC2 cells[22,34]. This difference in phasing between cytosolic and membrane rhythms may generate differential calcium and voltage relationships that reflect population-specific inputs and outputs.

The importance of the VIP/VPAC2 serial connection is evident when the link is broken in mice lacking genes encoding either VIP or VPAC2, with disrupted network oscillation and circadian behaviour[35,36]. VIP and VPAC2 cells are not, however, equal contributors, because ablation of VIP cells reduced amplitude and disrupted ensemble synchrony, whereas loss of VPAC2 cells only affected amplitude. This suggests a model whereby VIP cells synchronise the rest of the SCN, including VPAC2 cells, while non-VPAC2 cells (including VIP cells) can synchronise in the absence of VPAC2-expressing cells. The mechanism mediating VPAC2-independent synchronisation awaits clarification, although it may involve Prok2, GRP and GABA signalling[11,18,20]. In addition, astrocytes, which are active when SCN neurons are quiescent[19], have recently been shown to play a central role in driving the SCN circuit[26,37] and may well contribute to the spatio-temporal wave. Ultimately, understanding SCN circuit topology will inform analysis of the origin of the wave, while more detailed knowledge of wave structure will highlight underlying circuit topology.

VIP cells mediate retinal entrainment and thereby ensemble phase. Correspondingly, their optogenetic activation, alone or as part of the DRD1A population, can reset the SCN slice in culture[13] (and this study) and the mouse in vivo[38], while their inhibition blocks light-induced resetting in vivo[39]. These effects must involve trans-synaptic activation of target cells, which was evidenced directly as Per2::Luciferase induction in SCN regions outside the VIP$^{ChR2::EYFP}$ core. Moreover, optogenetic resetting of the SCN slice is attenuated by VPAC2 antagonism[13], and so resetting may act through the same VIP-to-VPAC2 pathway that directs the spatio-temporal wave. It is striking, however, that optogenetically increasing the firing rate of VPAC2 cells did not,

of itself, control ensemble phase, even though exogenous VIP stimulates SCN firing[40]. The phase-setting effect of VIP cells must therefore be via additional effects on VPAC2 cells, for example induction of CRE-dependent transcription, which in turn regulates their cell-autonomous TTFL[15]. Ultimately, changes in phase of VPAC2 cells, some of which express AVP and/or Prok2, will be transmitted to the rest of the SCN network and distal targets, via paracrine and/or autocrine signalling[18,29].

Disruption of network function allows cell-autonomous oscillations to continue but their periods rapidly diverge[6,8], suggesting that ensemble period is determined by circuit-wide computations[2]. Identification of which cells set ensemble pace has relied on intersectional approaches to create temporally chimeric SCN whereby the cell-autonomous period of a defined subset of SCN neurons has been genetically altered. Two partially overlapping cell populations expressing NMS[16] or DRD1A[17] have been shown to control SCN circadian period as a function of their own TTFL period. Both populations are, however, heterogeneous containing AVP, VIP and other cells. Selective targetting of VIP cells alone had no effect on behavioural circadian period[16], while AVP cells can impose their period on behaviour but not the isolated SCN slice[41,42]. Here we show, using Cry1-complementation, that VIP-to-VPAC2-mediated synchronisation draws all SCN cells to a common period, and VIP and VPAC2 cells together were able to impose their cell-autonomous period on the SCN almost as effectively as when all cells were complemented with Cry1[27]. This was also evident in vivo, further demonstrating the power of the VIP/VPAC2 cellular axis. The effectiveness of DRD1A and NMS cells in setting ensemble period may therefore be because they contain the VIP and VPAC2 populations. Such re-assignment of the VIP/VPAC2 cellular axis as the effective pace-setter is consistent with the disruption and/or short-period oscillations seen in VIP-null mice[36] and SCN[43]. In contrast, deletion of NMS is without effect on circadian timekeeping[16]: NMS signalling per se, therefore, is unlikely to contribute significantly to circuit computations mediated by VIP and VPAC2 cells.

Finally, the definitive emergent property is de novo oscillation. In contrast to commonly adopted loss-of-function approaches which test the necessity of cell-autonomous clocks in cell groups[16,41], we used Cry1-complementation of Cry1,2-null SCN to test for their sufficiency. Pan-neuronal expression of Cry1 initiated de novo circadian oscillations in Cry1,2-null SCN[26,28] (and current study), but restricted expression in either VIP cells or VPAC2 was ineffective. Notably, joint expression of Cry1 in VIP and VPAC2 cells rapidly initiated molecular cycles with appropriate ensemble period, cellular synchronisation and phase divergence. This circuit-wide effect was driven by Cry1 expression in ca. 50% of SCN cells. It is therefore possible that neither VIP cells nor VPAC2 cells alone were sufficiently abundant to exert control. Against this quantitative interpretation is the observation that with untargetted Cry1 expression, transduction of between 10 and 35% of neurons is sufficient to initiate robust behavioural rhythms in Cry-null mice[26,28]. It appears, therefore, that the intrinsic properties rather than the abundance of the VIP and VPAC2 cells are critical to circuit-level organisation of the SCN. Moreover, this organisation was readily transmitted to the control of circadian behaviour.

The current study has shown that neurons within the neurochemically explicit and topologically defined VIP-to-VPAC2 axis are able to define the emergent network-level properties essential to the role of the SCN as the circadian pacemaker, in vitro and in vivo. This network is not, however, restricted to neurons: SCN astrocytes also have a cell-autonomous TTFL, are able to set circadian period[19,37] and maintain SCN network function (albeit less potently than neurons)[26]. The challenge, now, is to understand how these individual elements, neuronal and astrocytic,

assemble into a coherent, multi-layered network[2]. The current study provides a platform onto which we can assign circadian properties to functional SCN units with the ambition to generate a complete topological network map. Linking network topology, function and emergent properties in the spatiotemporal domain in this way will have important implications not only for our understanding of the circadian clock, but also across other circuits of the brain[44].

## Methods

**Mice.** All experiments were carried out in accordance with the UK Animals (Scientific Procedures) Act of 1986, with local ethical approval provided by the Medical Research Council Laboratory of Molecular Biology Animal Welfare Ethical Review Board (LMB AWERB) and overseen institutionally by designated animal welfare officers (NACWOs). VIP$^{Cre}$ mice (Viptm1(cre)Zjh/J) were purchased from the Jackson Laboratory (Bar Harbor, Maine, USA). VPAC2$^{Cre}$ mice (Tg(Vipr2-cre) KE2Gsat/Mmucd) were purchased from GENSAT (Gene Expression in the Nervous System Atlas) project (Rockefeller University, New York City, USA). R26 floxed STOP EYFP (B6.129×1-Gt(ROSA)26Sortm1(EYFP)Cos/J) mice were kindly provided by A. McKenzie, MRC Laboratory of Molecular Biology, Cambridge, UK. Cry1-null and Cry2-null animals were derived from founders kindly provided by G. van der Horst, Erasmus University Medical Centre, Rotterdam, NL. Per2:: Luciferase mice were kindly provided by J. S. Takahashi, University of Texas Southwestern Medical Centre, Dallas, USA. All lines were maintained on a C57BL/ 6J background with required genotypes bred in-house by intercrossing the lines.

**AAVs.** AAVs used were a mix of commercially available AAVs and custom made AAVs. The following AAVs were sourced directly from Penn Vector Core and via Addgene: Cre-conditional tdTomato (AAV1-CAG-FLEX-tdTomato-WPRE (Addgene viral prep # 51503-AAV1)), pan-neuronal GCaMP6f (AAV1.Syn. GCaMP6f.WPRE.SV40 (Addgene viral prep #100837-AAV1)), Cre-conditional neuronal GCaMP6f (AAV1.Syn.Flex.GCaMP6f.WPRE.SV40 (Addgene viral prep # 100833-AAV1)), pan-neuronal ArcLight (AAV1.hSynap.ArcLightD.WPRE.SV40 (Addgene viral prep # 100037-AAV1)), Cre-conditional neuronal ArcLight (AAV1. hSynap.Flex.ArcLightDco.WPRE.SV40 (Addgene viral prep # 100038-AAV1)), pan-neuronal ChR2::EYFP (AAV1.Syn.hChR2(H134R)-EYFP (Addgene viral prep # 26973-AAV1)), Cre-conditional ChR2::EYFP (AAV1.EF1a.double floxed.hChR2 (H134R)-EYFP.WPRE.HGHpA (Addgene viral prep # 20298-AAV1)), Cre-conditional mCherry (AAV8-hSyn-DIO-mCherry (Addgene viral prep # 50459-AAV8)) and Cre-conditional EYFP (AAV1-EF1a-DIO-EYFP (Addgene viral prep # 27056-AAV1)). pAAV-CAG-FLEX-tdTomato-WPRE was a gift from Hongkui Zeng. pAAV.Syn.GCaMP6f.WPRE.SV40 and pAAV.Syn.Flex.GCaMP6f. WPRE.SV40 were gifts from Douglas Kim & Genie Project[45]. pAAV.hSynap. ArcLightD.WPRE.SV40 and pAAV.hSynap.ArcLightDco.WPRE.SV40 were gifts from Vincent Pieribone. pAAV.hSyn.hChR2(H134R).EYFP, pAAV.EF1a.double floxed.hChR2(H134R)-EYFP.WPRE.HGHpA and pAAV.EF1a.DIO-EYFP were gifts from Karl Deisseroth. pAAV-hSyn-DIO-mCherry was a gift from Bryan Roth. The following AAVs backbones were produced in-house and packaged by Penn Vector Core as AAV1 serotype: Cre-conditional pCry1-luciferase (pCry1.DIO.luc) and Cre-conditional Cry1::EGFP (pCry1.DIO.Cry1::EGFP).

**Cloning, AAV production.** For packaging into AAV vectors, the DtR-containing transgenic cassette was cloned into pAAV-EF1a-double floxed-hChR2(H134R)-mCherry-WPRE-HGHpA (Addgene #20297). Gibson cloning was used to insert an mCherry fluorescent protein and the simian DtR separated by a P2A peptide (to generate separate proteins in equimolar amounts) between the four loxP sites contained within the plasmid. hChR2-mCherry was excised from pAAV-EF1a-double floxed-hChR2(H134R)-mCherry-WPRE-HGHpA using BsrGI and NheI, linearising it in the process. mCherry-P2A was amplified from HIV.CMV.mCherry-P2A-Cre using the forward primer 5′-TAACTTCGTA TAGGATACTTTA-TACGAAGTTATGCTAGCCACCatggtgagcaagggcgagg-3′ and the reverse primer 5′-GCTTCATagggcgggattctcctccacgtc-3′ (capitalised letters represent regions of the primers complementary to the vector backbone and the DtR sequence respectively). The DtR sequence was amplified from pAAV.pCAG.DIO.DtR.WPRE, kindly donated by Dr. Marco Tripodi (MRC Laboratory of Molecular Biology, Cambridge, UK) using the forward primer 5′-GTGGAGGAGAATCCCGGCCC Tatga agctgctgccgtcgg-3′ and the reverse primer 5-CATTATACGAAGTTATG GCG CGCCTTACTTGTACAtcagtgggaattagtcatgcccaacttc-3′ (capitalised letters represent regions of the primers complementary to the P2A sequence and the vector backbone respectively).

The resulting PCR fragments were purified and incubated with the linearised vector backbone and Gibson mix (NEB) for 30 min at 50 °C. The reaction was chilled on ice before adding 5 µl of the mix to chemically competent cells, kindly donated by Dr. Ernesto Ciabatti (MRC Laboratory of Molecular Biology, Cambridge, UK). This was left on ice for 2 min and then plated onto LB plates containing ampicillin. Colonies were selected, mini-prepped and confirmed by sequencing. Following validation in cells, the Ef1a.DiO.mCherry-P2A-DtR plasmid was packaged into AAV1 serotype vectors by Penn Vector Core.

**SCN explant culture, bioluminescence and fluorescence imaging**. Mice (P8-10) were sacrificed according to local and Home Office rules, and the suprachiasmatic nucleus (SCN) was removed and cultured as an explant. Briefly, coronal hypothalamic slices were cut at 300 μm and the SCN was dissected free using a razor blade in ice-cold GBSS supplemented with (in mM): 5 mg/ml glucose, 50 μM D-AP5, 100 nM MK-801 and 3 mM MgCl₂. Slices were maintained in the interface method for 2–3 h in media containing: 50% Eagle's Basal Medium (Gibco), 25% EBSS (Gibco), and 25% Horse Serum supplemented with 5 mg/ml Glucose, 2 mM GlutaMAX (Gibco), 1:100 dilution of Penicillin/Streptomycin (Gibco), 50 μM D-AP5, 100 nM MK-801, and 3 mM MgCl₂. Following 2–3 h in culture, slices were incubated in the same media without the addition of D-AP5, MK-801 and MgCl₂ for a week. After a week, culture medium was changed, and 1-μl AAVs (between 1 × 10¹² and 1 × 10¹³ GC/ml in PBS) were added drop-wise to the surface of the slice 24 h later. Transduced slices were left for one week before AAVs were washed out by fresh culture medium and, in most cases, successful transduction was assessed by imaging.

For bioluminescent photomultiplier tube (PMT) recordings, slices were transferred to DMEM-based (Sigma-Aldrich) recording medium supplemented with: 4.17 mM NaHCO₃, 5 mg/ml glucose, 1:100 dilution of Penicillin/Streptomycin (Gibco), 10 mM HEPES, 5% FCS, 2 mM GlutaMAX, and 100 μM luciferin in 35-mm dishes. The dishes were then sealed with glass coverslips and vacuum grease before being transferred to a custom built PMT (H9319-11 photon counting head, Hamamatsu) array within a light-tight incubator at 37 °C. Bioluminescent emissions were collected in real time and binned into 6-min intervals before analysis. For bioluminescent imaging via CCD camera, slices were sealed into 35-mm dishes and transferred to the heated stage of an inverted microscope and CCD camera (Hamamatsu) setup. Bioluminescent time-lapse images were taken over 1-h intervals.

For combined bioluminescent and fluorescent imaging, slices were sealed into 35-mm dishes with glass bottoms (Mattek) and transferred to the heated stage of an LV200 microscope system (Olympus) running Olympus proprietary acquisition software (CellM, xcellence rt or cellSens) and equipped with an EM-CCD camera (Hamamatsu). Bioluminescence (PER2::Luciferase and pCry1-luc) and fluorescence (EYFP, GCaMP6f and ArcLight) images were taken once every 30 min, and recorded for at least 5 cycles. Exposure times ranged between 9.5 and 29.5 min for bioluminescence and 25 and 100 ms for fluorescent reporters (EYFP: 25–100 ms; GCaMP/ArcLight: 100 ms) dependent on the configuration of the experiment.

**Analysis of circadian imaging data**. Images were analysed in FIJI[46], IgorPro (Wavemetrics) using the SARFIA plugin, Excel (Microsoft) and Graphpad Prism (Graphpad). To generate phase maps and maps of bioluminescence/fluorescence intensity, slices were thresholded to remove extra-SCN signals before a custom FIJI plugin was used to extract signals from a continuous grid of ROIs placed over the SCN. In the case of phase maps, signals were detrended and smoothed by a 2.5 h moving average before peaks were identified and the peak-to-peak period of the PER2::LUC signal was used to calculate and normalise the phase of each cycle to allow the cycle to cycle and mean phase maps to be plotted for PER2 and GCaMP signals. For changes in normalised bioluminescence intensity, the change within each ROI was normalised to the baseline bioluminescence intensity within the same ROI. In order to define regions delineated by fluorescent markers, the average intensity projection was overlaid with the same grid and the intensity of the signal was normalised to the mean intensity across all ROIs. Areas with a normalised intensity of <1 were defined as outside the region, while areas with a normalised intensity of >1 were defined as within the region. Continuous phase maps were assembled and coloured in Graphpad Prism using the heat-map plot and were coloured using the "rainbow" colour setting. In the case of cluster phase-maps, phases were classified by manually assigning colours to any ROIs falling within the following phase ranges: <CT10, CT10–11, CT11–12, CT12–13, CT13-14 and >CT14.

CoL was measured using IgorPro scripts utilising the SARFIA toolset. CoL trajectory area was calculated from the perimeter of the CoL excursion and expressed as a percentage of the area of the temporally integrated bioluminescence signal. Circadian bioluminescence and fluorescence signals were analysed using a manual peak identification and peak-to-peak period approach in the case of phase-mapping electrophysiology experiments (Fig. 2), fluorescence/bioluminescence experiments (Fig. 3) and optogenetics experiments (Fig. 6). Differences in phasing comparing pan-neuronal or targeted ArcLight and GCaMP6f were assessed statistically by ordinary one-way ANOVA with Holm-Sidak's correction for multiple comparisons. Potential differences in aggregate period between Per2::Luciferase and pan-neuronal or targeted (VIP^Cre and VPAC2^Cre) ArcLight/GCaMP6f were assessed statistically by paired t-tests. Differences in phasing, Rayleigh vector length, CoL trajectory, baseline and amplitude for pCry1-luciferase rhythms were assessed by unpaired t-tests. Differences in period for pCry1-luciferase rhythms between pan-neuronal (synapsin-Cre) and targeted (VIP^Cre and VPAC2^Cre) conditions were assessed by ordinary one-way ANOVA with Holm-Sidak's correction for multiple comparisons. Bioluminescence recordings from slice cultures in long-term PMT recordings were analysed to calculate circadian period length, amplitude, relative amplitude error (RAE) and goodness of fit (GOF) using

the Fast Fourier Transform—Non-Linear Least Squares (FFT-NLLS) function in the BioDARE software[47] (www.biodare2.ed.ac.uk).

**Sectioning/immunostaining/confocal microscopy**. Mouse brains were post-fixed in 4% PFA (in phosphate buffer) and cryoprotected in 20% sucrose (in PBS) prior to sectioning. Brains were sectioned on a freezing microtome (Bright Instruments, UK) at 40-μm thickness. Sections containing the SCN were blocked with 5% NGS prior to immunostaining. Primary antisera used were: Rabbit anti-AVP (Peninsula, T-4563) (used at 1:1000 dilution), rabbit anti-VIP (Immunostar, 20077) (used at 1:750 dilution) and guinea pig anti-VIP (Peninsula T-5030) (used at 1:1000 dilution). Fixed brain sections and slices were imaged using the Zeiss 780 inverted system (Zeiss, Germany) with either ×10 or ×20 objective for low power images or ×63 objective for high power images, used for cell counting. Cell counting for co-localisation was done manually using the FIJI Nucleus Counter and Cell Counter plugins. Transduction efficiency of AAVs was calculated using AAV-dependent fluorescence cell counting in conjunction co-localisation. Efficiency of AAV1.pCry1.DIO.Cry1::EGFP transduction was expressed as a percentage of Cre+ cell counts, where Cre+ cells were revealed using AAV8.pSyn.DIO.mCherry transduction. AAV8.pSyn.DIO.mCherry transduction efficiency was then assessed by calculating the percentage of Cry1::EGFP-positive cells where mCherry is absent, termed "over-spill" expression.

**Riboprobe generation**. Due to spatial patterning of VIP-ir in the projections of VIP neurons, it can be technically difficult to perform cell counts. To confirm specificity of Cre-recombinase expression in the VIP-cells and perform cell counts, fluorescent in situ hybridisation (FISH) was performed against Cre-recombinase and VIP. Labelled riboprobes were generated via a one-step reverse-transcriptase PCR (RT-PCR) protocol using gene-specific primers to generate cDNA, followed by an in vitro transcription (IVT) to create the riboprobe. Primer sequences were sourced from the Allen Brain Atlas (VIP forward: CCTGGCATTCCTGATACT CTTC, VIP reverse: ATTCTCTGATTTCAGCTCTGCC; Cre forward: CCAATT TACTGACCGTACACCA, Cre reverse: TATTTACATTGGTCCAGCCACC) and synthesised (Sigma) containing either the bacterial promoter T7 sequence (TAA TACGACTCACTATAGGGAGA; forward primer) or T3 sequence (AATTAAC CCTCACTAAAGGGAGA; reverse primer) 5′ to the rest of the sequence to generate sense and antisense probes, respectively, via IVT. A one-step reverse-transcriptase (RT)-PCR was carried out (SuperScript III One-Step RT-PCR System with Platinum *Taq* kit, Invitrogen) to reverse-transcribe RNA extracted from SCN slices (RNeasy Micro, Qiagen) and amplify the resultant cDNA. 50–100 ng RNA was used as a template. Once the resultant cDNA was purified (QIAquick PCR Purification kit, Qiagen), 500 ng was in vitro transcribed using either T3 or T7 polymerase (Roche), in a reaction mix containing 1X transcription buffer (Roche), 0.01 M DTT, digoxigenin (DIG) or fluorescein (FLU) labelling mix (Roche), RNasin, and polymerase. This was left at 37 °C for 2 h in a thermocycler and was subsequently cleaned up using Micro Bio-Spin 6 columns (Bio-Rad) before diluting 1:5 in nuclease-free water and storing at −20 °C.

**Fluorescent in situ hybridisation (FISH)**. Mice were culled by cervical dislocation and exsanguination, and dissected brains were immediately frozen on aluminium foil on top of dry ice, SCN facing upwards. Twelve μm sections were taken using a cryostat and allowed to air dry. These sections were post-fixed in 4% PFA made up in DEPC-PBS for 20 min at 4 °C. Following three washes in DEPC-PBS, sections were incubated in triethanolamine (TEA; Sigma) solution (1.4 M TEA in DEPC-H₂O) for 3 min at room temperature before being acetylated in 0.25% acetic anhydride in TEA solution for 10 min at room temperature. Sections were subsequently washed in DEPC-PBS before dehydration by ascending concentrations of ethanol (50, 70, 95 and 100%) for 3 min each. Sections were then left to air-dry before addition of hybridisation buffer (62.5% deionised formamide, 375 mM NaCl, 1.25x Denhardt's solution, 12.5 mM Tris (pH 8), 1.25 mM EDTA, 12.5% dextran sulphate, supplemented with 250 μg/ml Torula yeast RNA and 12.5 mM dithiothreitol.

RNA probes were diluted 1:100 in hybridisation buffer, heated at 60 °C for 10 min and then immediately chilled on ice for 5 min. Eighty microlitres of probe solution was added to each slide and a coverslip was placed on top of the solution to ensure uniform spreading and reduce evaporation. Slides were placed in a hybridisation chamber (saturated with 50% formamide and 50% PBS) and incubated overnight in a ventilated oven at 58 °C. Following hybridisation, coverslips were removed by washing in 4× saline-sodium citrate buffer (SSC) before treating with 20 μg/ml RNase A for 30 min at 37 °C. The sections were subsequently washed in decreasing SSC concentrations (2×, 1×, 0.5×) before a final wash in 0.1x SSC at 60 °C. Sections were then washed and equilibrated in Tris-NaCl buffer (TNB; 0.1 M Tris (pH 8), 150 mM NaCl) before blocking in 5% normal sheep serum (in TNB) for 30 min. The primary antibody (anti-DIG-POD) was diluted 1:100 in this blocking buffer and 80 μl of this antibody solution to each slide, cover-slipping and incubating at 4 °C overnight. Sections were washed three times in TNB to remove the antibody solution and were then treated with TSA working solution (TSA Plus Cy3 diluted in TSA Diluent 1:50; Perkin Elmer): 80 μl was added to each slide, which were then cover-slipped and incubated in the dark at room temperature for 1 h to produce a fluorescent precipitate. Three washes in

TNB followed before incubating the slides in 3% $H_2O_2$ (in TNB) for 30 min. Sections were again blocked and treated with the remaining antibody (anti-FLU-POD) at a 1:100 dilution and incubated overnight as before. The washes and TSA treatment were also repeated, using TSA plus FLU. Three washes in TNB and a final rinse in water were carried out before immediately cover-slipping with Vectashield Hardset mounting medium with DAPI.

**Electrophysiology**. For electrophysiological recordings, SCN slices were maintained in PMTs and their bioluminescent output was monitored to determine phase and extrapolate time-stamped recordings in circadian time. Slices were transferred to the stage of an upright (BX51WI, Olympus) microscope by excision of the supporting membrane and maintained in ACSF (in mM: 125 NaCl, 25 NaHCO3, 1.25 NaH2PO4, 3 KCl, 25 Glucose, 2 CaCl2, 1 MgCl2) bubbled with 95% CO2/5% O2, constantly perfused at ~1.5 ml/min at 32 °C via an inline heater. For cell-attached recordings, borosilicate glass pipettes (3-6MOhms) were filled with 150 mM NaCl and recordings were made in voltage clamp mode using a Cornerstone BVC-700A amplifier (Dagan). Signals were digitised at 10 kHz and filtered at 1 kHz. For electrophysiological optogenetic characterisation, cells recorded in cell-attached voltage clamp mode were identified by their ChR2::EYFP tag. Each cell recorded was stimulated with a train of 60 equally spaced pulses (15 ms) delivered at 1, 5, 10, 15 and 20 Hz using a computer-controlled TTL-triggered LED array (pE-4000, 470 nm LED, CoolLED) via the widefield standard EGFP fluorescence cube. Firing index was calculated by dividing the number of evoked spikes by 60 (the number of light pulses delivered), with values of <1 representing failure of the stimulation pattern to induce firing at the desired rate.

For whole-cell recordings, borosilicate glass pipettes (4–8 MOhms) were filled with a K-gluconate based internal solution (in mM: 135 K-gluconate, 7 NaCl, 10 HEPES, 2 Na2-ATP, 0.3 Na2-GTP, 0.01 Biocytin-Alexa488, 2 MgCl2) and recordings were made using a Cornerstone BVC-700A amplifier (Dagan). Targeted recordings were made via visualisation of the tdTomato signal excited using widefield illumination via LED illumination (pE-4000, 550 nm LED, CoolLED) via a standard Texas Red fluorescence filter cube. Signals were digitised at 20 kHz and filtered at 3 kHz. Series resistance for whole-cell recordings was monitored at break in and at the end of the recording, and if it changed by >25%, recordings were omitted. Input resistance was measured by applying a series of 500 ms long hyperpolarising current steps (−30pA to 0pA, 10pA steps) to neurons and measuring the steady-state voltage deflection. Input resistance was defined as the gradient of the linear relationship between the voltage deflection and the injected current. All recordings were completed within 3–4 min of break-in to minimise effects of intracellular perfusion, but cells were allowed to perfuse for up to 10 min to confirm successful targeting by registration of the green Alexa-488 signal with the red tdTomato signal and images were acquired using micromanager (ImageJ). For all whole-cell recordings, correction of the junction potential was applied a posteriori.

Analysis was performed in AxoGraphX (AxoGraph), Igor Pro (Wavemetrics) using the Neuromatic plugin[48], Excel (Microsoft), Graphpad Prism (Graphpad) and R (version 3.6.1, courtesy of R Foundation for Statistical Computing, Vienna, Austria. https://www.R-project.org) using RStudio (version 1.2.1335, courtesy RStudio Team, Boston, MA. http://www.rstudio.com). The regularity of firing was assessed quantitatively using the LvR metric (reproduced from ref. [25]):

$$\text{LvR} = \frac{3}{n-1}\sum_{i=1}^{n-1}\left(1 - \frac{4I_iI_{i+1}}{(I_i+I_{i+1})^2}\right)\left(1 + \frac{4R}{I_i+I_{i+1}}\right),\qquad(1)$$

where $I_i$ and $I_{i+1}$ represent consecutive interspike intervals (ISIs), $n$ is the number of ISIs and $R$ represents the refractoriness constant (defined as 5 ms[25]) to correct for the refractory period within an ISI directly following a spike and define regularity as a variable independent of firing rate.

For pooled and time-aligned electrophysiological data, a linear mixed model was fitted to the data to take into account differences in the numbers for neurons recorded between slices. For pooled electrophysiological data, to examine the underlying differences between the two genotypes (VIP[Cre] or VPAC2[Cre]), the linear mixed model was fit using genotype as a fixed factor with slice as a random factor. In the time-aligned datasets, the linear mixed model was fit using genotype, time and an interaction between the two as fixed factors with slice as a random factor. The models were fit using the nlme package in R (nlme version 3.1.143) (courtesy R Core Team nlme: https://CRAN.R-project.org/package=nlme). In the case of the pooled electrophysiological data, the significance of the genotype effect was assessed by applying an unpaired, two-tailed Welch's t-test. For the time-aligned data, the significance of the fixed factors in the model were determined through application of a two-way ANOVA. The two-way ANOVA was followed by pair-wise comparisons made on the estimated marginal means derived from the model and Tukey's correction was applied to these multiple comparisons made using the emmeans package in R (emmeans version 1.4.3.1) (Estimated Marginal Means, aka Least-Squares Means. R package version 1.4.3.01. https://CRAN.R-project.org/package=emmeans).

**Optogenetics**. SCN slices were recorded in specially adapted PMTs with fibre optic cables running into the base of the chamber to allow for optogenetic stimulation (1 h, 10 Hz) in situ via computer-controlled LEDs (in-house hardware

and software, Electronics Workshop, MRC Laboratory of Molecular Biology). Stimulation strength of both LED wavelengths (470 nm: M470F, Thor Labs/625 nm: M625F, Thor Labs) was measured to be at least 1 mW/mm². Slices were recorded for 4 cycles before stimulation and this was used to extrapolate the phase of stimulation and to predict the peaks post-stimulation. Slices were recorded for 4 cycles post-stimulation and phase shifts were calculated as the difference between the actual peak and the predicted peak normalised in circadian time to the baseline period and expressed as the mean of the three cycles following stimulation. For the Cry-null optogenetic experiments, stimulation was given after recording a 24 h stable PER2::Luciferase baseline. In the case of direct imaging, slices were checked for a stable baseline in PMTs before being transferred to an LV200 bioluminescence/fluorescence imaging system (Olympus). Bioluminescence and EYFP signals were captured every 30 min before optogenetic stimulation (1 h, 10 Hz) was delivered to the slice via the GFP cube. Slices were then recorded for a further 24 h. Significance was assessed in the PRCs via two-way ANOVA with Sidak's correction for multiple comparisons. Bioluminescent changes in Cry-null SCN receiving pan-neuronal or VIP-targeted optogenetic stimulation were analysed via paired t-tests or repeated-measures two-way ANOVAs with Sidak's correction for multiple comparisons.

**Toxin ablation**. Following baseline recording of SCN slices for >5 days, Dtx or vehicle (recording media) was added to the bulk media. Dtx (Sigma) was bath-applied to SCN slices at a final concentration of 20 ng/ml. Following 5–7 days of treatment, Dtx or vehicle was washed off via media change and slices were recorded further for >5 days. Recordings were analysed in BioDare, values were calculated based on rhythms 4 days pre-treatment and 4 days post-treatment excepting the first 36 h immediately after treatment application. Data were analysed via either a two-way ANOVA or three-way ANOVA, followed by Sidak's correction for multiple comparisons.

**AAV-mediated Cry1 complementation**. Cry1-null SCN slices were recorded in PMTs for 7 days to confirm arrhythmicity. They were then transduced with 1 μL of AAV.pCry1.DIO.Cry1EGFP and returned to the PMTs for a further 7 days, followed by a media change and return to PMTs for a final 7 days. Pre- and post-analysis of Per2::Luciferase rhythms used the first and last 7 days respectively. In order to follow transient period changes from cycle to cycle as Cry1::EGFP expression increased following transduction, peak-to-peak period was calculated using the area under a curve feature in Graphpad Prism to automatically identify the time of PER2::LUC peaks. Time intervals between these automatically identified peaks were expressed as the period for that cycle. Period comparisons were made via a repeated-measures two-way ANOVAs with Sidak's correction for multiple comparisons. Cry1,2-null SCN slices were recorded from in PMTs for 7 days to confirm arrhythmicity. They were then transduced with 1 μL of AAV.pCry1.DIO.Cry1::EGFP and returned to the PMTs for a further 7 days, followed by a media change and return to PMTs for a final 7 days. Pre- and post-analysis of Per2::Luciferase rhythms used the first and last 7 days respectively. After PMT recordings, slices were imaged on CCD camera. Finally, the slices were transduced with AAV8.Syn1.DIO.mCherry to confirm Cre-recombinase expression for one week before being fixed (4% PFA in phosphate buffer) and mounted using Vectashield with DAPI (Vectorlabs, U.S.A) on slides ready for confocal imaging. Cell counting was performed in FIJI using the "nucleus counter" plugin. %EGFP cells was calculated relative to DAPI cell counts. Data were analysed using BioDARE[49]. Comparisons of period between pan-neuronal and VIP[Cre]/VPAC2[Cre] initiations were made via unpaired two-tailed t-tests. GOF measures were compared via a repeated-measures two-way ANOVA with Sidak's correction for multiple comparisons. Trajectory area and Rayleigh vector data was compared by ordinary one-way ANOVA with Kruskal–Wallis correction for multiple comparisons.

**In vivo stereotaxic injection of AAVs**. The sex of SCN slices was not known. Therefore, both male and female adult (aged between 10 to 14 weeks) mice expressing both VIP[Cre] and VPAC2[Cre] on a Cry1,2-null background from two separate cohorts ($n = 4$ male, 4 female; $N = 8$) were individually housed in cages equipped with running wheels with food and water available ad libitum. As part of the breeding strategy, Cry1-null, Cry2-heterozygous animals were also produced and subjected to the same stereotaxic surgery to determine the in vivo pacemaking efficiency the VIP and VPAC2 cells ($n = 3$ male, 2 female; $N = 5$). Mice were singly caged in cages equipped with running wheels with environmental conditions complying with NC3Rs guidelines of ambient temperatures of between 20–24 °C and relative humidity between 45 and 65%. Mice were entrained to a 12-h light:12-h dark schedule (lights on at 07:00) for ca. 10 days, followed by 10 days in continuous darkness (DD) to assess their endogenous free-running period. The mice then received bilateral stereotaxic injections of AAV1 pCry1-DIO-Cry1::EGFP (0.3 μl per site) into the region of the SCN (±0.25 mm medio-lateral to Bregma, 5.5 mm deep to dural surface) under halothane anaesthesia. Following surgery, the mice were maintained on a 12-h light:12-h dark schedule for several days for recovery before transferring back to DD for re-assessment of their endogenous behavioural period. To confirm that administration of AAV particles to the SCN led to expression in the VIP and VPAC2 cell populations within the SCN, mice were culled and brain harvested, fixed in 4% PFA, cryo-preserved overnight in 20%

sucrose and sectioned (40 μm) on a sledge freezing-microtome. Confocal microscopy of native EGFP signal was used to identify successful targeting of the SCN. Due to inefficient targeting, two animals were removed from further analysis in the Cry1,2-null cohort, and three animals were removed from the Cry1-null, Cry2-heterozygous cohort.

Behavioural data were analysed using ClockLab (Actimetrics, Inc., USA) running within Matlab (Mathworks, USA). Activity profiles were generated by outputting the last 7-days of recordings from ClockLab binned at 12-min intervals across a 27 h day (the average post-surgery period) for both pre-and post surgery. Profiles were then averaged to generate a single 27 h profile for each animal which was then normalised to the highest peak in the activity trace. Traces were then aligned so that their activity onsets were centered at the mid-point of the 27 h day, and averaged across all animals to generate an average activity profile.

For Cry1,2-null cohorts, paired comparisons were made between pre- and post-surgery conditions using a paired one-tailed Student's t-test. A one-tailed test was deemed appropriate due to a priori assumptions that can be made from the ex vivo SCN data. Comparisons were made between slice and behavioural data using a two-tailed unpaired t-test. For Cry1-null,Cry2-heterozygous cohort, comparisons were not made against slice data because the final number of animals (n = 2) was deemed too low for the statistical analysis to be reliable.

**Reporting summary**. Further information on research design is available in the Nature Research Reporting Summary linked to this article.

## Data availability
The source data underlying Figs. 1d, f, 2b–g, j, k, 3e, g, i, 4g, 5c, e–h, k, l, 6a–d, 7b–f, and 8c–f and the source data underlying Supplementary Figs. 4c, d, 5b, d, e, h, I, 6c–e, 7d–j, 8a, 9b–d, 11b and 12b are provided as a Source Data file. Other data and biological materials are available from the corresponding author on reasonable request.

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

## Acknowledgements

The authors would like to acknowledge the invaluable support of the MRC Laboratory of Molecular Biology Electronics and Mechanical Workshops, and Ares (Biological Services Group). This work is funded by MRC core funding (MC_U105170643) and BBSRC grants (BB/R016658/1 and BB/P017355/1).

## Author contributions

A.P.P., M.D.E., N.J.S., R.H. and M.H.H. designed the project. A.P.P. and M.H.H. wrote the manuscript. A.P.P. performed the electrophysiology, calcium and voltage imaging, spatiotemporal analysis and drafted the figures. A.P.P and M.D.E. performed the optogenetic experiments. N.J.S. and M.D.E. performed transcriptional *pCry1*-luc and Cry1 complementation experiments. R.H. performed immunohistochemistry, in situ hybridisation and diptheria toxin ablation experiments. E.S.M. and J.E.C. performed in vivo behavioural experiments. J.E.C. supervised mouse husbandry. M.B. performed pre-liminary experiments. M.H.H. supervised the study. All authors provided feedback on the manuscript.

## Competing interests

The authors declare no competing interests.
