## [Peer Review File · Nature Communications]

Reviewer #2 (Remarks to the Author)

This manuscript extends the multi-faceted analysis of SCN physiology. It addresses the network properties of this complex nucleus in particular as the smaller components generate an ensemble rhythm in each of several properties (TTFL, calcium and voltage). As I understand it, the main question posed by this paper is summarized here:

Line 123 "Put simply, which cells and signalling axes are the pacemakers of the SCN? To address this, we focussed on the neurochemically explicit VIP/VPAC2 axis for reasons outlined above^{12,13}, and the observation that paracrine VIP-ergic signalling can re-programme circuit function^{20,14,15}."

My principal question concerns the overall novelty of the result presented. It is not clear to me what is the main advance in our understanding of SCN network physiology. The wave of TTFL activities across the SCN and the importance of VIP- and VPAC2-neurons have been well-described. So how precisely do the present results extend previous findings, and what substantial new findings are now brought forward for a general audience?

Line 184 The SCN-wide emergent wave of $[Ca^{2+}]_i$ therefore contains locally specific sub-waves attributable to VIP and VPAC2 cells.

These interesting observations require better definition in the narrative: a wave is set to be phase-advanced relative to another – in the text, please specify the difference in hours or minutes. It is shown in figures, but not in text. Presumably, these phase differences are critical for this study, have substantial biological meaning and so should be detailed in the text. Other descriptions of regional waves would likewise benefit by such quantitative specificity in the text.

Line 445 "voltage across the network has been reported as simultaneous²¹." Meijer and colleagues (Vanderleest 2007 Current Biol) reported that activity of individual SCN neurons is limited to a few hours and staggered across the day and night, with more events in the day. Is there reason to favor certain reports over others?

Minor questions/ comments:

Line 113 "highlighting potential topological features of the SCN circuit that include several neuropeptidergic signalling axes."

What are neuropeptidergic signaling axes? – these sound significant but the implied meaning is obscure. Aren't all neuronal classes signaling axes?

Line 92 "Importantly, although synchronised, the phases of cellular TTFLs across

the SCN are staggered as localised cell groups exhibit peak circadian activities in a distinct sequence.”

Given the ensemble rhythm, what is the significance of these phase differences? How does that translate to different outputs?

Line 152 “Regional distribution of VIPCre and VPAC2Cre cells, as revealed by Cre-dependent fluorescence from the calcium reporter GCaMP6f (delivered by adeno-associated virus (AAV)), mapped to different phase-clusters:”

GCaMP6 is not an obvious choice to map spatial differences as it is an activity-dependent signal. Normally, membrane-bound reporter proteins (nominally “inert”) are used for spatial determinations

Line 135 “As anticipated, EYFP labelled cells exhibited contrasting SCN distributions: VIPEYFP cells clustered ventrally within the core and VPAC2EYFP cells localised around the dorsal, lateral and medial shell”(Supplementary Fig. 1a,b).

Not clear why this was anticipated, as there are reports that SCN cells broadly express VPAC2. Is this contrast supported by SCN cell profiling?

Fig 2K – a ~1 hr phase advance in peak depolarization was measured between two cell populations using Arclight. However, I could not find any description of Arclight methods – how often, how long, or when?

How was efficacy of DTX assessed? In addition, what orientation was used for hypothalamic slices? De la Iglesia and Schwartz reported differences in phasic PER-LUC activity according to the orientation of hypothalamic slices – one orientation revealed

Line 464 “The mechanism mediating VPAC2-independent synchronisation awaits clarification, although it may involve other topological elements including Prok2, GRP and GABA signalling^{20,18,11.}”

I believe the text should read ‘VPAC2 cell-independent synchronization’.

Reviewer's comments.

All textual edits are tracked into the main text and legends and methods in **green highlight** and reproduced below.

Reviewer #2 (Remarks to the Author):

a. General point on novelty.

*My principal question concerns the **overall novelty of the result** presented. It is not clear to me **what is the main advance** in our understanding of SCN network physiology.*

To perform its role, the SCN has to generate a powerful and coherent, circuit-based circadian time signal. The cellular origins of the emergent, circuit-level properties that confer this power and coherence are not known and are an area of considerable interest. In our view, the best approach to date has come from the Takahashi laboratory who made it clear that the cell-autonomous circadian properties of NMS-expressing cells dictate SCN circuit function (oscillation and ensemble period) (Lee et al. Neuron, 2015). The drawbacks of this work are, first, that this cell population is very abundant across the SCN and incorporates many other cell-types, and so the manipulations lack neurochemical specificity. Second, loss of the ligand NMS has no impact on the circuit or behaviour and so the relevance of this peptidergic axis is questionable. We therefore “flipped” this approach by analysing a system in which we already knew the ligand and receptor are important, in order to ask whether and how the cell-autonomous properties of the respective cell populations contributed to circuit function: are these neurochemically defined SCN cell populations pacemakers?

The main advances are:

1. We provide an in-depth characterisation of the respective electrophysiological and molecular (gene expression, Ca²⁺ levels) circadian properties of VIP- and VPAC2-expressing cells. This allowed us to reveal their distinct features and, in so doing, to map them onto the behaviour of the SCN circuit. We also demonstrated their respective contribution to local “waves” of cellular activity: an unprecedented finding (see below Reviewer's comments re. waves). This cell-type-specific differential characterisation and mapping to reveal the origin of ensemble properties has not before been achieved. No-one has explored the circadian properties of VPAC2 cells in this way, nor have any pairs of explicitly defined cell-types (ligand- and receptor-expressing) been compared directly in this way.

2. We identify these VIP- and VPAC2-expressing cells as a pacemaking axis of the SCN circuit by showing that their cell-autonomous circadian properties together dictate the emergent properties of the circuit. Although VIP and VPAC2 have been shown to be important as molecules in “holding” the circuit together, the explicit roles of the cells that express them have not been shown before, and there is no *a priori* reason to assume that these functions would overlap. Knowledge of one gene product is not knowledge of a cell's function.

Specifically, we show that by working together, but not individually, the VIP- and the VPAC2-expressing cell populations can determine the period of the clock, imposing their cell-autonomous period on the other circadian-competent cells embedded in the

circuit. We further show that the cell-autonomous clocks of VIP- and VPAC2-expressing cell populations, together, can initiate circadian function in an otherwise circadian-incompetent SCN (again with a period determined by those cells). These properties of the VIP- and VPAC2-expressing cells have not been shown before, and emphasise that the work is novel.

3. The multi-plexed imaging procedures and genetic manipulations we have applied are state-of-the-art, and incorporate a number of novel features that, we anticipate, will be of broad interest and utility.

b. Specific point on TTFL “waves”

The wave of TTFL activities across the SCN and the importance of VIP- and VPAC2-neurons have been well-described. So how precisely do the present results extend previous findings, and what substantial new findings are now brought forward for a general audience?

Line 184 The SCN-wide emergent wave of [Ca²⁺]_i therefore contains locally specific sub-waves attributable to VIP and VPAC2 cells.

These interesting observations require better definition in the narrative: a wave is set to be phase-advanced relative to another – in the text, please specify the difference in hours or minutes. It is shown in figures, but not in text. Presumably, these phases differences are critical for this study, have substantial biological meaning and so should be detailed in the text. Other descriptions of regional waves would likewise benefit by such quantitative specificity in the text

We are not sure how to proceed here because first of all the Reviewer seems to question the novelty of the findings, and then s/he goes on to agree with us, saying that we should make more of them. In particular, it seems that we all consider our discoveries regarding global and local waves to be of interest. We have therefore amended the text as requested by the Reviewer, providing the explicit phases of calcium, electrical activity, and transcriptional rhythms to make more explicit our findings, as follows:

“Importantly, when registered against the Per2::Luciferase rhythm, the cycle of VIP^{GCaMP6f} cells was significantly phase-advanced relative to pan-neuronal and VPAC2^{GCaMP6f} oscillations (Fig. 1f) (Neuronal: CT 7.6±0.2h; VIP: CT6.1±0.4h; VPAC2: CT7.8±0.2h).”

“Importantly, the peak circadian phase was again significantly phase-advanced in VIP^{ArcLight} cells by ~1h relative to pan-neuronal and VPAC2^{ArcLight} cells (Fig. 2k) (Neuronal: CT 7.6± 0.2h; VIP: CT6.8 ±0.1h; VPAC2: CT7.4 ±0.2h).”

“Combined bioluminescent/fluorescent CCD recordings confirmed spatially appropriate Cry1-Luciferase signals, and phase-aligned bioluminescence curves revealed significant phase differences: VIP^{Cry1-Luc} cells peaking ~1.8h earlier than VPAC2^{Cry1-Luc} cells (VIP: CT11.0 ±0.2h; VPAC2: CT12.8 ±0.3h)”

c. Specific point on TTFL literature citation

Line 445 “voltage across the network has been reported as simultaneous²¹.” Meijer and colleagues (Vanderleest 2007 Current Biol) reported that activity of individual SCN neurons is limited to a few hours and staggered across the day and night, with more events in the day. Is there reason to favor certain reports over others?

We are pleased to have added Vanderleest et al. to our revision. This was easy to do because the paper supports our case, albeit using a different form of measurement (ephys vs. imaging). The original purpose of referring to reference 21, however, was that it used the same measurement as we did to monitor electrical activity, ArcLight, but it came to a different conclusion. We included it to highlight this discrepancy and to be sure that we did not ignore work that went against our view. On that specific, we consider the findings in reference 21 to be misleading because the ArcLight is expressed pan-neuronally at the membrane of neurons. As these neurons likely project widely across the SCN, the spatial localisation of the signal is washed out compared to cytosolic reporters (e.g. GCaMP) where the bulk of the cellular cytoplasm (and therefore signal) will be spatially restricted to the soma. Additionally, the phasing of these rhythms were assessed by looking at the phasing differences between calcium and voltage in arbitrarily specified ROIs, meaning that there is no standardisation of phasing between preparations to enable accurate comparisons to be made. In our approach, we demonstrate a difference in ArcLight phasing by genetically restricting the expression of ArcLight to the cell populations of interest and using bulk measurement of the Per2 and ArcLight signals to determine phase. This mitigates these issues by removing the need to show differences in phase spatially across the SCN (as the bulk ArcLight signal is genetically restricted), and the phase marker used to align recordings is consistent across preparations. We would add that whilst we disagree with the conclusion in reference 21 that electrical activity in the SCN is simultaneous, we agree that there is potentially a regional difference in phasing between calcium and voltage, perhaps underlying different inputs and outputs to those cells – something we make reference to in the discussion section. Despite using a different method to assess electrical activity, the findings in Vanderleest et al (now reference 12) chime perfectly well with our findings, and we now write:

“In the current study, we utilised two functionally relevant components (VIP and VPAC2) to delineate core and shell to reveal differential phasing between these two compartments, an observation that is consistent with reported phase-staggering of the electrical activities of individual neurons across the circadian cycle¹².”

d. Minor questions/ comments:

Line 113 “highlighting potential topological features of the SCN circuit that include several neuropeptidergic signalling axes.”

What are neuropeptidergic signaling axes? – these sound significant but the implied meaning is obscure. Aren't all neuronal classes signaling axes?

Yes, we over-worded it. What we meant by “axis” was to focus on populations of cells explicitly linked by the expression of ligand and receptor, analogous to the networks inferred by Park et al (Frontiers in Neuroscience, 2016, doi: 10.3389/fnins.2016.00481). We have re-worded this to explicitly define this concept as follows:

“In addition, cells expressing neuromedin S (NMS) and D1a dopamine receptors (DRD1A) extend across core and shell^{17,18}, whilst single-cell transcriptomics has revealed further cellular heterogeneity, highlighting potential topological features of the SCN circuit that include several signalling axes consisting of interacting neuropeptide- and receptor-expressing cellular populations¹⁹.”

Line 92 “Importantly, although synchronised, the phases of cellular TTFLs across the SCN are staggered as localised cell groups exhibit peak circadian activities in a distinct sequence.”

Given the ensemble rhythm, what is the significance of these phase differences? How does that translate to different outputs?

We do not know and we would love to know their significance. As discussed by us and by others, the distributed phases that arise from distinct cell-autonomous properties (n.b. a similar calcium-reported differential phasing/ activity wave is seen in the clock circuitry of *Drosophila*) may be a design feature of the circuit that stabilises oscillation. Such an emergent feature is observed in mathematical models of coupled oscillators. More directly biological, it may well be that differential phasing enables the SCN to convey distinct time-stamped activational and/ or inhibitory signals to its various targets. Finally, many groups have considered the possibility that phase spread is important in encoding daylength. One motivation to our current work is that if we are successful and can identify which cells become active in which order, we can then, in future, target them selectively in order to test the above hypotheses. The current findings showing distinct temporal properties of VIP- and VPAC2-expressing cells and establishing the principle that these cells have the ability to control the SCN circuit (and now in the revision, circadian behaviour) augur well for this ambition. Such work is, however, beyond the scope of the current studies.

Line 152 “Regional distribution of VIP^{Cre} and VPAC2^{Cre} cells, as revealed by Cre-dependent fluorescence from the calcium reporter GCaMP6f (delivered by adeno-associated virus (AAV)), mapped to different phase-clusters:”

GCaMP6 is not an obvious choice to map spatial differences as it is an activity-dependent signal. Normally, membrane-bound reporter proteins (nominally “inert”) are used for spatial determinations.

Even though in inactive cells there is a baseline expression of fluorescent signal, this is a fair point and so we have repeated the analysis using Cre-dependent expression of EYFP. The results are comparable to those with GCaMP but with finer resolution and have now replaced the data in Figure 1a with additional data being presented in Supplementary Figure 3. We have amended the text as follows:

“Regional distribution of VIP^{Cre} and VPAC2^{Cre} cells, as revealed by Cre-dependent EYFP fluorescence (delivered by adeno-associated virus (AAV)), mapped to different phase-clusters: VIP^{EYFP} neurons were within the ventral phase-advanced core, straddling regions 2 to 4, and with a ventral-to-dorsal progression (Fig. 1a, Supplementary Fig. 3a). Conversely, the VPAC2^{EYFP} neurons mapped dorsally across the shell, straddling delayed phase-regions 4 to 6, with a predominantly lateral progression (Fig. 1a, Supplementary Fig. 3b). This coarse mapping suggests that the

two cell-types are differentially phased with contrasting spatiotemporal dynamics of circadian activation.”

Line 135 “As anticipated, EYFP labelled cells exhibited contrasting SCN distributions: VIP-EYFP cells clustered ventrally within the core and VPAC2-EYFP cells localised around the dorsal, lateral and medial shell” (Supplementary Fig. 1a,b).

Not clear why this was anticipated, as there are reports that SCN cells broadly express VPAC2. Is this contrast supported by SCN cell profiling?

a. SCN cells broadly express VPAC2

It is certainly the case that SCN cells broadly express *Vipr2*, by our estimate ca. 38%, but overlap with VIP-expressing cells was low. When considering cell-type expression, it is important to recognise that the methods used to co-visualise mRNA and peptides/proteins have different relative merits. The evidence for co-expression of VPAC2 by VIP-expressing cells comes predominantly from An et al. (J. comp. Neurol. (2012) 520:2730–2741). Immunostaining on the intact SCN sections in that study, however, lacked the resolution to identify cellular co-expression: both VIP-ir and VPAC2-ir signals are highly diffuse across the SCN and the mice were not treated with colchicine prior to sampling to restrict expression to cell bodies. The explicit evidence for co-expression in An et al. comes not from imaging the intact nucleus, but from dual-immunostaining of dispersed cultures of neonatal SCN. The distribution in dispersed culture may well be prone to artefactual mis-expression of either or both antigens. Moreover, co-localisation was represented by 3 SCN cells in a single figure, lacking quantitative analysis. The representative merit of the qualitative analysis is therefore unclear. On the other hand, using in situ hybridisation to map the expression of *Vip* and *Vipr2* mRNA provides a more definitive answer regarding the distribution of cell-types. We therefore anticipated a contrasting core vs. shell distribution on the basis of published work and our own unpublished in situ hybridisation studies, as well as the corresponding profiles from the Allen Brain Atlas.

b. Is this contrast supported by SCN cell profiling?

On the specific of SCN single cell profiling, Park et al. (Frontiers in Neuroscience, 2016, doi: 10.3389/fnins.2016.00481) provided explicit evidence for separate expression of these two genes, with no evidence for autocrine signalling by VIP onto VPAC2-expressing VIP cells. This further informed our anticipation. Furthermore, in a recently released study, Wen et al. (Nature Neuroscience, 2020, doi: 10.1038/s41593-020-0586-x) also show that VIP and VPAC2 cells are spatially and neurochemically segregated through smFISH and single cell profiling.

c. Not clear why this was anticipated

Perhaps we should not have said “anticipated” and simply reported our direct observations, which we now do. It is not controversial to report that *Vip* and *Vipr2* mRNA are not co-expressed by SCN cells: *Vip*-expressing cells are in the SCN core and *Vipr2* cells are in the shell.

The text is amended accordingly:

“EYFP-labelled cells exhibited contrasting SCN distributions: VIP^{EYFP} cells clustered ventrally within the core whereas VPAC2^{EYFP} cells localised around the dorsal, lateral and medial shell (Supplementary Fig. 1a,b). Immunostaining and in situ hybridisation of SCN sections showed clear segregation of VIP and VPAC2 (or AVP) cells (Supplementary Fig. 1c-f), with minimal evidence for mutual VIP-VIP cell signalling via VPAC2. These lines therefore provided selective genetic access to distinct VIP or VPAC2 cellular compartments.”

Fig 2K – a ~1 hr phase advance in peak depolarization was measured between two cell populations using ArcLight. However, I could not find any description of ArcLight methods – how often, how long, or when?

The description was present, but was compressed alongside other AAV-delivered reporters. We have now amended the text in the materials and methods section as follows:

“For combined bioluminescence (PER2::Luciferase and pCry1-luc) and fluorescence (EYFP, GCaMP6f and ArcLight) imaging, bioluminescence and fluorescence images were taken once every 30 minutes, and recorded for at least 5 cycles. Exposure times ranged between 9.5 and 29.5 minutes for bioluminescence and 25 and 100 milliseconds for fluorescent reporters (EYFP: 25-100ms; GCaMP/ArcLight: 100ms) dependent on the configuration.”

How was efficacy of DTX assessed?

We counted the number of cells expressing Cre-dependent EYFP after Cre-dependent Dtx treatment, finding that the Dtx treatment resulted in a >99% reduction in DtR-expressing cells (as assessed by mCherry and EYFP signals). The remaining Cre-expressing cells revealed by the post-treatment super-transduction of slices with the Cre-dependent EYFP AAV revealed that the relative inefficiency of DtR expression was <10%. We have added these cell counts to Supplementary Figure 7 and have amended the text to reflect this as follows:

“SCN slices transduced with DtR highlighted VIP^{Cre} or VPAC2^{Cre} cells with spatially appropriate mCherry signals, >99% of which were lost following specific ablation by addition of diphtheria toxin (Dtx) when compared with vehicle controls (Supplementary Fig. 7a-d).”

In addition, what orientation was used for hypothalamic slices?

All our work was conducted in the coronal plane, the standard view for such imaging studies and so directly comparable to many (the majority) other reports.

De la Iglesia and Schwartz reported differences in phasic PER-LUC activity according to the orientation of hypothalamic slices – one orientation revealed..... (There may be a typo deletion in this comment).

The paper which we think is referred to (Jagota et al., Nature Neuroscience (2000), PMID 10725927) was conducted in Syrian hamster SCN slices and it monitored electrophysiological activity, revealing a unitary phase of activity in coronal slices and a bimodal pattern, with “morning” and “evening” peaks, in horizontal slices. As reflected in the discussion above, the phase relationship of these peaks was photoperiodically sensitive, consistent with the extreme photoperiodic sensitivity of behaviour and physiology of Syrian hamsters. Our work on mouse SCN always used animals reared on 12L::12D, a default condition but it is not unreasonable to suggest that if we worked with mice from extreme daylengths, the phase-spread we observed would be enhanced in the coronal plane and perhaps in the horizontal plane, provided we looked at it immediately after slice preparation. Indeed, work from the Evans/Davidson and Honma laboratories imaging gene expression on mouse SCN has shown as much. In our work, however the slices were left for some time (>2 weeks) to revert to their stable default condition. To seek to repeat all of the current analysis with slices prepared in the horizontal plane (and then why not sagittal plane also?) is beyond the scope of our study. Moreover, it would not invalidate our results from coronal slices.

Line 464 “The mechanism mediating VPAC2-independent synchronisation awaits clarification, although it may involve other topological elements including Prok2, GRP and GABA signalling^{20,18,11.}”

I believe the text should read ‘VPAC2 cell-independent synchronization’.

We agree and have corrected it: sorry. It is a tad ironic, maybe, but this comment makes the point to us that, after all, the Reviewer “gets” the idea that this study was about the cells and not about the molecules.

Reviewer #3 (Remarks to the Author):

The circadian timing output of the circadianin the generation of SCN network activity. The data provide important new insights into the organization and functioning of the SCN network and the role of two identified populations of SCN neurons.

Specific concerns:

1. In Figure 1a, the VIP^{Cre} cells are located in both regions three and four, suggesting that there is a phase dispersion of VIP neuronal activity.

Yes that is a fair interpretation of the data and it chimes with the observation we make later that there is a local spatiotemporal wave in conditionally restricted GCaMP6f and pCry1-luc reports (Fig. 1g, Supplementary Fig. 4e-g, Fig. 3h and Supplementary Fig. 6g-h). We have amended the text to make a clearer reference to the fact that this is suggestive of a wave in gene expression across these populations:

“Regional distribution of VIP^{Cre} and VPAC2^{Cre} cells, as revealed by Cre-dependent EYFP fluorescence (delivered by adeno-associated virus (AAV)), mapped to different phase-clusters: VIP^{EYFP} neurons were within the ventral phase-advanced core, straddling regions 2 to 4, and with a ventral-to-dorsal progression (Fig. 1a, Supplementary Fig. 3a). Conversely, the VPAC2^{EYFP} neurons mapped dorsally across the shell, straddling delayed phase-regions 4 to 6, with a predominantly lateral progression (Fig. 1a, Supplementary Fig. 3b). This coarse mapping suggests that the two cell-types are differentially phased with contrasting spatiotemporal dynamics of circadian activation.”

2. The authors' interpretation of the ArcLight recordings compared to the single neuron membrane recordings is equivocal. The single electrode recordings are a direct measure of the membrane potential, while the ArcLight recordings will reflect a population response. The rhythm in ArcLight recordings may reflect not the resting membrane potential, but the fact more neurons are firing at higher frequencies.

Yes, again we agree. We cannot prove this using our current approaches and ArcLight as an electrophysiological tool. We were equivocating the ArcLight signal to resting membrane potential, as this is an established use previously by papers utilising ArcLight to make optical recordings of electrical activity in the SCN (Brancaccio et al, 2017; Enoki et al, 2017). However, as we cannot adequately address this concern, we have altered the text to express the ArcLight report as a measure of electrical activity rather than membrane potential and have removed the explicit reference to the single electrode recordings:

“To obtain complementary analyses of circuit-level electrophysiological activity, we transduced SCN with the voltage sensor AAV synapsin-ArcLight, expressed either pan-neuronally^{20,23} (Supplementary Fig. 5f-h) or conditionally in VIP or VPAC2 cells (Fig. 2h), and phased-registered the fluorescence to simultaneous whole-field Per2::Luciferase recordings. Both VIP^{ArcLight} and VPAC2^{ArcLight} neurons demonstrated

stable circadian rhythms in electrical activity (Fig. 2i,j). Importantly, the peak circadian phase was again significantly phase-advanced in VIP^{ArcLight} cells by ~1h relative to pan-neuronal and VPAC2^{ArcLight} cells (Fig. 2k) (Neuronal: CT 7.6± 0.2h; VIP: CT6.8±0.1h; VPAC2: CT7.4±0.2h). Thus, VIP and VPAC2 cells are electrophysiologically distinct and exhibit differentially phased (VIP advanced) circadian electrophysiological rhythms.”

3. There is no quantification of the efficiency of the AAV transfections and subsequent expression of the different molecular tools used in the study.

Figure 7d did present quantification of the number of the AAV-transduced cells, but not the efficiency of transduction. To answer this point, we have conducted new experiments in which we focussed on characterising the most functionally relevant tool whereby we could contrast phenotype with targeting efficiency with a high degree of sensitivity – Cre-dependent Cry1 expression in Cryptochrome-null SCN slices. In these experiments, we characterised the targeting efficiency of AAV1 pCry1-DIO-CRY1::EGFP by super-transducing slices with AAV8 Syn-DIO-mCherry. We rationalised that the mCherry report will provide a measure of the efficiency of the Cre recombinase, and the Cry1 expression would provide a measure of the efficiency of AAV mediated Cry1 restoration. Post experiment imaging has enabled us to demonstrate that the efficiency of targeting is ca. 90% (mCherry positive cells that are also EGFP positive, Supplementary Figure 9b,d) across all three genotypes (VIP-Cre, VPAC2-Cre and VIP;VPAC2-Cre) with an “EGFP overspill” of <10% (EGFP positive cells that are mCherry negative, Supplementary Figure 9c) which likely represents the inefficiency of the mCherry AAV. Additionally, we show that the efficiency of the transduction does functionally alter our results – where the Cre-recombinase is targeted determines the initiation, not the number of cells targeted or how efficiently those cells are targeted (Figure 7d, Supplementary Figure 9d). Thus, using two different AAVs with different promoters and different serotypes (which represent the predominant serotypes in this study), we have identified that our AAV transductions combined with the Cre-recombinase expressing mouse lines have a targeting efficiency in excess of ca. 90%. As these experiments required us to obtain additional replicates to relate efficiency to function, these new data have been added to expand those already presented in Figure 7. In light of this improved assay of efficiency, we have replaced the old Supplementary Figure related to this data with a new Supplementary Figure using these new measures and have presented the efficiency measures therein (Supplementary Figure 9).

In addition, in answering *Reviewer 2*'s question of the efficiency of diphtheria toxin ablation, we have also identified that using this two AAV approach (Cre-dependent DtR-mCherry expression followed by super-transduction with Cre-dependent EYFP), we are able to obtain similar numbers for efficiency for this assay: ca. 90% efficiency of mCherry or EYFP expression with a “spill over” (inefficiency) of ca. 10%.

4. Alternate statistical tests need to be used in several of the experiments (for example - Fig. 2 and Supplemental Figure 4). The authors averaged the values for all the neurons in a single slice then used the mean values in their statistical calculations. This approach does not take into account the number of neurons studied and potential different variances between slices. It also gives extra weight to

the neurons recorded from slices with fewer recorded neurons. A linear mixed model that takes into account repeated measures of a single neuron and multiple neurons from each slice is one test that could be used.

We are pleased to say that we have applied a linear mixed model to the data as requested, and the outcome does not alter our interpretation of the data from our previous “mean of mean” approach. We have incorporated this statistical testing into the relevant figures (Fig. 2 and Supplementary Figure 5) and have updated the methods section accordingly:

“For pooled and time-aligned electrophysiological data, a linear mixed model was fitted to the data to take into account differences in the numbers for neurons recorded between slices. For pooled electrophysiological data, to examine the underlying differences between the two genotypes (VIP-Cre or VPAC2-Cre), the linear mixed model was fit using genotype as a fixed factor with slice as a random factor. In the time-aligned datasets, the linear mixed model was fit using genotype, time and an interaction between the two as fixed factors with slice as a random factor. The models were fit using the nlme package in R (nlme version 3.1.143) (courtesy R Core Team nlme: <https://CRAN.R-project.org/package=nlme>). In the case of the pooled electrophysiological data, the significance of the genotype effect was assessed by applying an unpaired, two-tailed Welch’s t-test. For the time-aligned data, the significance of the fixed factors in the model were determined through application of a two-way ANOVA. The two-way ANOVA was followed by pair-wise comparisons made on the estimated marginal means derived from the model and Tukey’s correction was applied to these multiple comparisons made using the emmeans package in R (emmeans version 1.4.3.1) (Estimated Marginal Means, aka Least-Squares Means. R package version 1.4.3.01. <https://CRAN.R-project.org/package=emmeans>).”

Reviewers' Comments:

Reviewer #2:

Remarks to the Author:

I am satisfied with the responses provided by the authors.

They have provided additional clarification as well as specific additional measures to bolster their interpretations.

I have no further concerns.

Reviewer #3:

Remarks to the Author:

Multiple studies have identified VIP and VPAC2 expressing neurons as critical for the functioning of the suprachiasmatic nucleus and the generation of circadian timing signals. The current manuscript examines the role of VIP and VPAC2 expressing neurons in the regulation of SCN network activity over the circadian cycle. The data demonstrate that these two populations of neurons together control rhythmicity. The data presented provides additional details on the functioning of the SCN neural network. The authors have been very responsive to the first review and answered my initial concerns. In particular, the electrophysiological data was analyzed with the appropriate statistical model. The authors should be explicit in the Discussion that both the VIP expressing cells (Kawamoto et al., 2003) and the VPAC2 expressing cells represent diverse neuronal populations. The two neuronal populations make up a significant proportion (52%, Figure 9) of the total number of neurons in the SCN, raising a question of whether some of the effects are due to the disruption of a significant percentage of SCN neurons.

Response to Reviewers

Reviewer #2 (Remarks to the Author):

I am satisfied with the responses provided by the authors.

They have provided additional clarification as well as specific additional measures to bolster their interpretations.

I have no further concerns.

We thank the Reviewer for their comments, and acknowledge that the resulting changes have substantially improved the manuscript.

Reviewer #3 (Remarks to the Author):

Multiple studies have identified VIP and VPAC2 expressing neurons as critical for the functioning of the suprachiasmatic nucleus and the generation of circadian timing signals. The current manuscript examines the role of VIP and VPAC2 expressing neurons in the regulation of SCN network activity over the circadian cycle. The data demonstrate that these two populations of neurons together control rhythmicity. The data presented provides additional details on the functioning of the SCN neural network. The authors have been very responsive to the first review and answered my initial concerns. In particular, the electrophysiological data was analyzed with the appropriate statistical model.

We again thank the Reviewer for their comments, and acknowledge that those comments have helped us to improve the manuscript substantially.

1. The authors should be explicit in the Discussion that both the VIP expressing cells (Kawamoto et al., 2003) and the VPAC2 expressing cells represent diverse neuronal populations.

We agree. We have added the following at the start of the Discussion, which includes the Kawamoto et al. Reference.

“...VIP and VPAC2 cells represent diverse neuronal populations^{32, 33} and they have distinct electrical signatures....”

2. The two neuronal populations make up a significant proportion (52%, Figure 9) of the total number of neurons in the SCN, raising a question of whether some of the effects are due to the disruption of a significant percentage of SCN neurons.

The Reviewer raises a salient point: are the effects of any one sub-population a function of the intrinsic qualities of that population, or a consequence of its overall abundance, i.e. a quantitative effect. We note in the text how the literature has provided evidence for both perspectives, especially where targeting of SCN cells has not been neurochemically specific. With regards to the VIP and VPAC2 cells, we feel that this is addressed in Figure 7d, where the relative percentages of AAV-recruited cells in the “Cry1 initiation” experiment are shown versus the goodness-of-fit (GOF) a measure of the robustness of the initiated rhythm (where a lower GOF value indicates a more robust rhythm). This panel shows that even though comparable numbers of VPAC2 cells or VIP plus VPAC2 cells can be recruited as a function of AAV-transduction in different slices, it is only when the VIP and VPAC2 cells are recruited together that a robust circadian oscillation is initiated. These results therefore show that, for this signalling axis at least, the type of cells recruited determines outcome, hence our statement: “...Rather, it was due to the specific recruitment of both cellular constituents of the VIP neuropeptidergic axis.”